**SOFTWARE**

# PEGR: a flexible management platform for reproducible epigenomic and genomic research

Danying Shao[1], Gretta D. Kellogg[2], Ali Nematbakhsh[2], Prashant K. Kuntala[3], Shaun Mahony[3], B. Franklin Pugh[4] and William K. M. Lai[4,5*]

*Correspondence:
wkl29@cornell.edu
[5] Department
of Computational Biology,
Cornell University, Ithaca, NY
14850, USA
Full list of author information
is available at the end of the
article

## Abstract

Reproducibility is a significant challenge in (epi)genomic research due to the complexity of experiments composed of traditional biochemistry and informatics. Recent advances have exacerbated this as high-throughput sequencing data is generated at an unprecedented pace. Here, we report the development of a *P*latform for *E*pi-*G*enomic *R*esearch (PEGR), a web-based project management platform that tracks and quality controls experiments from conception to publication-ready figures, compatible with multiple assays and bioinformatic pipelines. It supports rigor and reproducibility for biochemists working at the bench, while fully supporting reproducibility and reliability for bioinformaticians through integration with the Galaxy platform.

**Keywords:** Genomics, Galaxy, High-throughput sequencing, Data management system, Reproducibility, Science gateway

## Background

Reproducibility is one of the cornerstones of scientific research. However, reproducibility has been a longtime challenge across many scientific fields [1–5]. These difficulties arise from complexities in experimental and bioinformatic workflows that diverge over time, across different operators, and often with limited versioning [6–9]. In the field of genomics, collections of massive datasets that can be parsed in many ways have added to the reproducibility challenge [10–17]. One method to address these issues is to apply systematic metadata capture and management software that is tailored to (epi)genomic data collection.

 In general, a genomic project is composed of two distinct but interrelated components: "wet-bench" biochemistry experiments and "dry-bench" bioinformatic analysis. In wet-bench experiments: sample type (human tissue biopsy, yeast, etc.), reagents (catalog number, wash buffer recipe, etc.), growth environment (log growth, % confluence,

etc.), and experimental protocols (ChIP-seq, Western blot, etc.) are examples of critical metadata that need to be captured. Minor variations in these experimental components can result in distinct experimental outcomes [18, 19]. Confounding these issues is the traditional reliance on storing experiment metadata in hand-written notebooks, which are not searchable and often incomprehensible to a third party [20]. Consequently, it can be difficult to follow and accurately reproduce an experimental protocol from start to finish.

Similarly, in bioinformatics analysis, different analytical tools, software versions, and tool parameters may generate different analytical outcomes. While progress has been made in tracking and reproducing informatic workflows (e.g., Pegasus, Galaxy), these platforms are generally limited to reproducing software workflows [21, 22]. To our knowledge, there are no free open-source platforms in active development that manage entire experimental pipelines, from wet-bench experiments to bioinformatic analyses [23]. Laboratory information management systems (LIMS) typically focus on inventory management and sample tracking and have limited capability to record experimental metadata, data analysis parameters, and for interfacing with project team members [24–26]. Although there have been several commercial efforts in this direction, they can be limited in scope (e.g., only tracking sequencing reagents) and/or rather expensive to small academic laboratories [27, 28]. These platforms typically have limited integration between data production and a wide range of experimental metadata.

We developed the *P*latform for *E*pigenomic and *G*enomic *R*esearch (PEGR), a web-based project management platform that integrates wet-bench sample tracking with downstream bioinformatic workflows (managed by Galaxy workflows) to address the challenge of experimental reproducibility. PEGR provides end-to-end management of (epi)genomics projects from initial reagent usage through to the reproducible generation of publication-quality figures. PEGR logs sample information and experimental details. It manages metadata produced by Galaxy or any other workflow system and provides sample reporting and visualization. It supports Findable, Accessible, Interoperable and Reusable (FAIR) best practices by tracking the metadata throughout the entire sequencing pipeline to enable experimental reproducibility [29]. Previously, we presented a proof-of-concept vision of PEGR software [30]. We now present a fully functional and open-source version of PEGR software, including streamlined experiment tracking with reagent barcoding, flexibility for multiple heterogeneous bioinformatics workflows, and a project management approach. To date, PEGR has been officially acknowledged in two papers generating thousands of genomic datasets and is actively utilized by multiple (Lai, Pugh, and Mahony) research labs and the Cornell Epigenomic Core Facility [31, 32]. PEGR is freely available and open source at https://github.com/seqcode/pegr.

## Results

### Overview

PEGR is a project management platform designed to organize, track, and disseminate the workflow of (epi)genomic projects from the start of an experiment through DNA sequencing, bioinformatic analyses, and figure generation (Fig. 1). It tracks sample information and sequencing metadata, manages the bioinformatics workflow, and

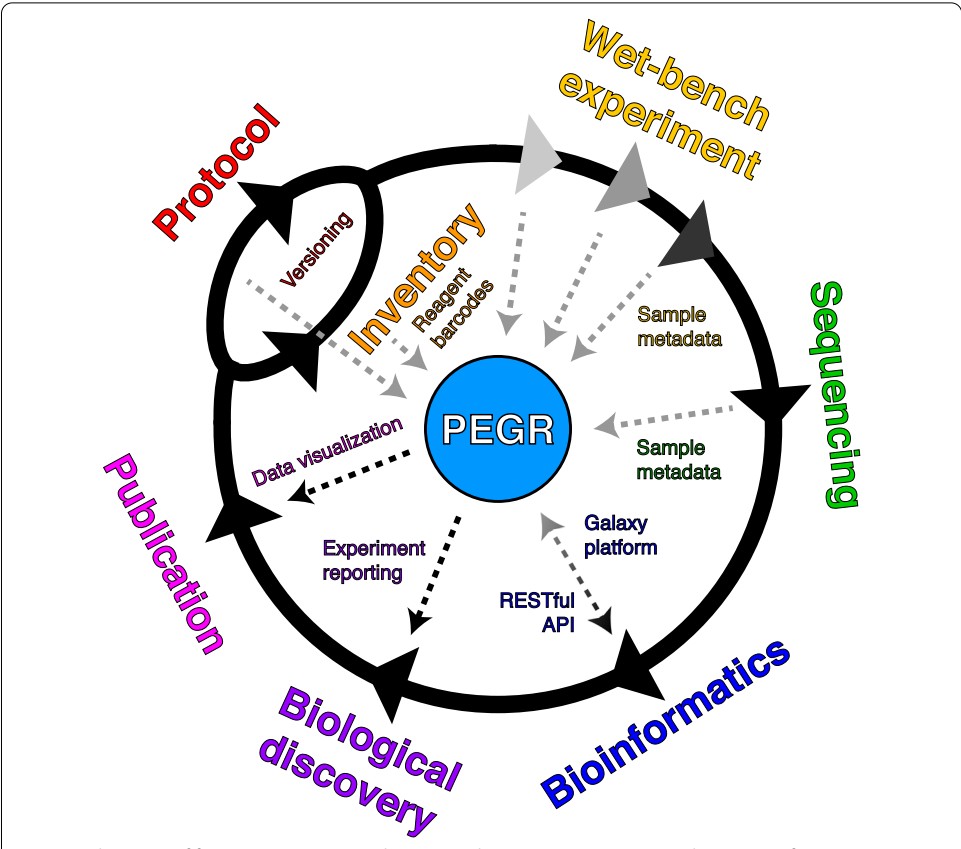

**Fig. 1** Schematic of feature integration within PEGR. The outer text represents the stages of an (epi) genomic workflow that PEGR manages. Arrows reflect workflow direction, whereas the inner text represents the conduits by which PEGR controls information flow through each stage. PEGR captures customizable metadata based upon the settings and fields set by an administrator for the platform

provides QC reporting and visualization. PEGR is intended to enable a more complete scientific workflow from hypothesis generation through publication-quality figures.

PEGR supports user submission of detailed sample and experiment information through several methods, including a web-interface, real-time QR reagent barcode tracking with an Android app, and an Excel-based sample submission form. After experimental metadata has been recorded in PEGR (e.g., cell line, species, assay), a sequencing run can then be relationally linked to this information. PEGR tracks Illumina sequencing runs in real-time by periodically probing the sequencer's output data repository. When PEGR detects the completion of a sequencing run (i.e., RunCompletionStatus.xml), it will automatically initiate an external bioinformatics workflow platform. Presently PEGR natively supports the Galaxy platform and is extensible to other workflow engine platforms such as Pegasus and bash shell scripting [22]. PEGR collects the metadata information of bioinformatic outputs in real-time and displays them in an online workflow monitoring dashboard. Users are also able to query PEGR programmatically using a RESTful API for metadata related to specific samples or workflow runs. Critically, PEGR maintains and tracks the relational links between the figures and analyses generated by the bioinformatic software back through the starting reagents for any given sample.

### Inventory management

PEGR provides an inventory management section in support of a key aspect of experimental reproducibility, creating the ability to simply and easily track the exact chemicals, enzymes, reagents, antibodies, equipment, and sample material that form the basis of experimental assays. While this form of reagent tracking is accomplished by most laboratories at a basic level, in practice, the quality of record-keeping can be variable depending on laboratory organizational structure and personnel training [20]. PEGR was architected to include an integrated inventory management system that seamlessly tracks all aspects of an experiment's metadata. To reduce incorrect and faulty information from being uploaded into PEGR, approved inventory entries (ItemType) can be predefined by administrative users (Fig. 2A). Metadata fields such as name, vendor, catalog

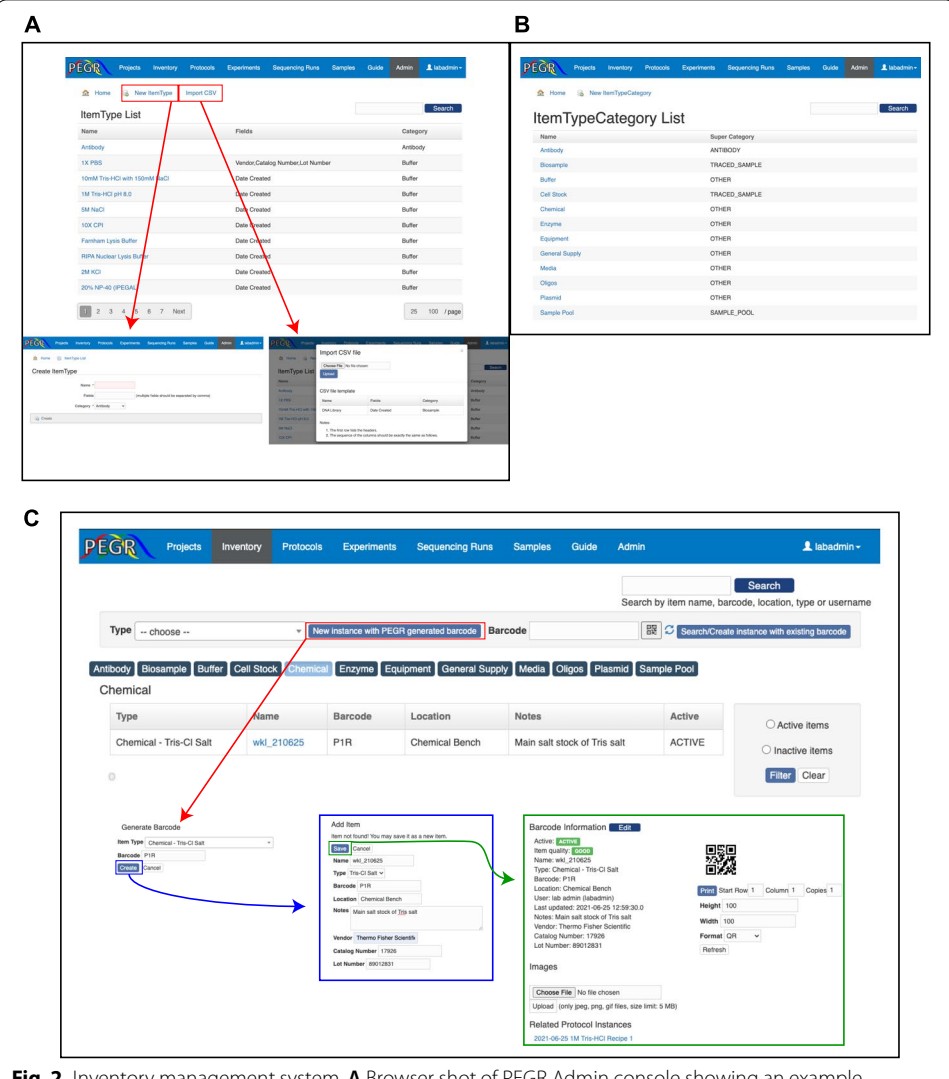

**Fig. 2** Inventory management system. **A** Browser shot of PEGR Admin console showing an example ItemType list. CSV files can be generated from existing lab management platforms such as iLab and Quartzy. **B** Browser shot of PEGR Admin console showing an example ItemTypeCategory list. **C** Example of instantiating a new ItemType in PEGR under the "Inventory" interface. Boxes and arrows from red to blue to green, highlight the steps taken in order, showing the links and windows that appear for generating a new barcode. QR barcode sizes are set in the web interface and are compatible with most label printers

number, and lot number can be added to each ItemType to help guide the initial deployment and match LIMS fields available for import.

PEGR was designed to provide maximum flexibility for an assay workflow. Custom fields can be defined for a specific item type in the admin console. Given the wide range of possible reagents and variables a lab may choose to track, PEGR provides a simple CSV upload form which allows an administrative user to upload a list of all inventory item types and the related aspects of inventory metadata that a laboratory desires to track (Fig. 2A). This provides compatibility with common laboratory management systems such as Agilent iLab and Quartzy which possess CSV export functionality of their tracked inventory [25, 26]. To prevent disorganization resulting from tracking all possible inventory items in a laboratory, ItemType's are grouped into an ItemTypeCategory (Fig. 2B). This structure allows for the intuitive organization of the full spectrum of a laboratory's inventory. The item types in the ItemTypeCategory list are defined through the web interface and provide the ability to dynamically assign and re-assign ItemTypes as the needs of the laboratory change.

The "Inventory" tab on the main PEGR navigation bar is the primary interface for tracking all instances of ItemTypes in a laboratory. New instances of ItemTypes can be created and old instances can be marked "inactive" by regular users (e.g., technicians, graduate students) as reagent stockpiles are finished (Fig. 2C). To reduce the "activation energy" required for adopting and maintaining in-depth tracking of reagent catalog numbers, specific lot numbers, aliquot dates, etc., PEGR leverages an easy-to-use QR barcode scanning application ("Barcode Scanner" app by ZXing) on Android devices that updates the PEGR backend database system in real-time [33]. The barcode scanner can be activated directly from a webpage in PEGR, and the result is returned to PEGR via a callback URL. Materials received by the lab that already have an attached barcode can be scanned from the Android devices and the appropriate metadata is recorded along with a time stamp. Purchased reagents and client samples with no existing barcode can be assigned a new barcode generated by PEGR. The barcode is shown in both text and 2D QR image in PEGR. PEGR's barcode system also integrates with existing label printers which allows for the 2D image to be printed in different sizes to accommodate the physical dimensions of the inventory.

### Experimental protocol versioning and integration with inventory management

PEGR provides a protocol assembly and management module. While the traditional method of protocol management for most wet laboratories is a physical paper binder containing common buffer recipes and basic experimental procedures, there are cloud-based approaches for experimental protocol management such as OpenWetware (https://openwetware.org) and Protocol-Online (http://www.protocol-online.org/). In contrast to these approaches, PEGR's protocol management system directly links its laboratory inventory metainformation with tracked and version-controlled experimental protocols (Fig. 3A).

Defining the exact ItemType input and output for each protocol is crucial for PEGR to properly track laboratory metadata across an experiment. When new experiments are initialized, the user is required to follow the predefined protocol and record the items used during an experimental setup. It is required that all the item types defined

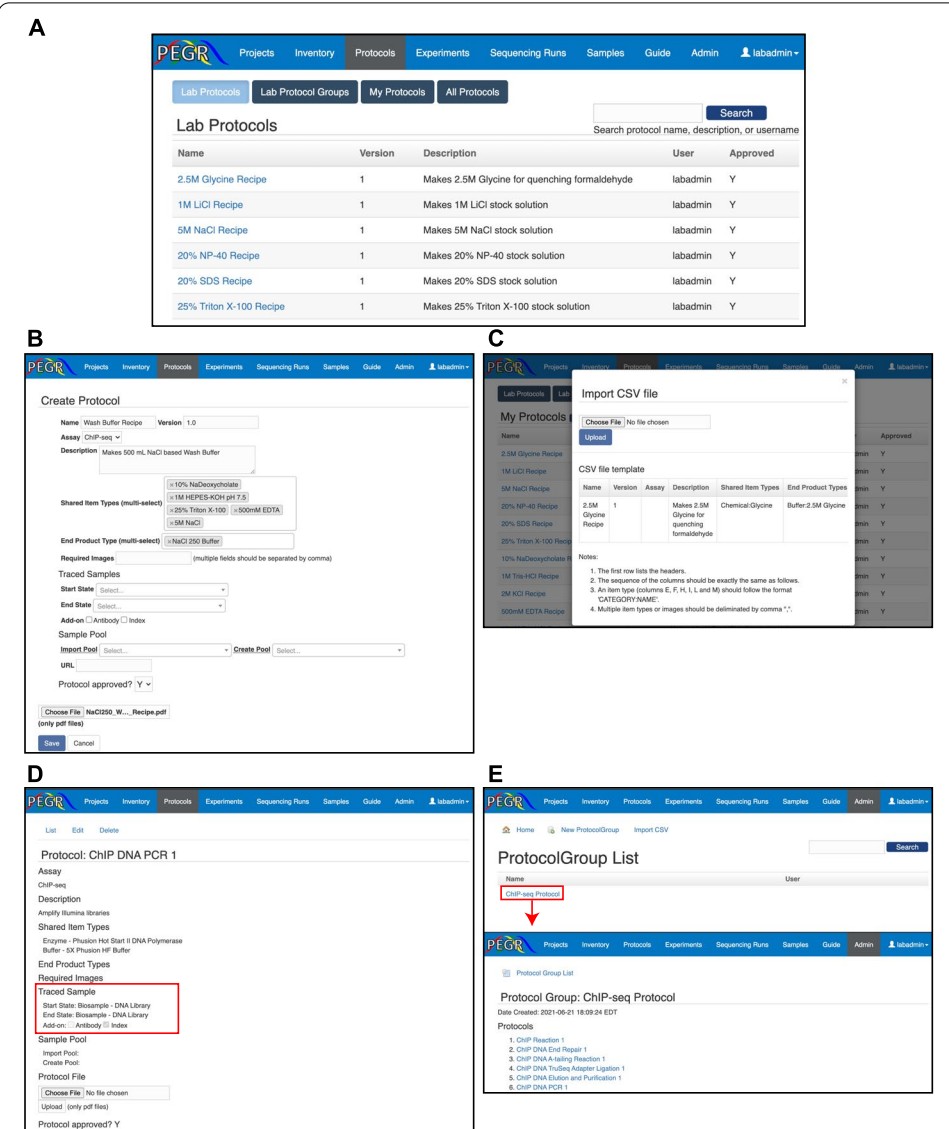

**Fig. 3** Protocol management system. **A** Browser shot of PEGR "Protocols" interface. Default view displays protocols that have been approved by an administrator. Personal (nonapproved) protocols are available under "My Protocols". **B** Browser shot of PEGR webform to create a new protocol. PEGR inventory database autocompletes fields where possible in Shared Item Types (input) and End Product Type (output) fields. **C** A CSV webform to upload protocols in bulk to the PEGR "Protocol" interface. A template guide appears to assist upload. **D** Traced samples are defined in a "Lab Protocol" and represent a sample that undergoes changes to its state (e.g., cell pellet into purified DNA). **E** "ProtocolGroup" is accessible through the Admin console and provides a webform and CSV option for defining new ProtocolGroups based on approved protocols

in the protocol be linked to an item instance before the user can move on to the next step. The process of defining a new protocol in PEGR is possible through two distinct options under PEGR's "Protocol" tab. The first method of generating a new protocol is directly through the PEGR user interface. A simple webform links to PEGR's ItemType database and allows for the creation of protocols ranging from simple buffer recipes to highly complex multi-stage assays such as ChIP-seq using a controlled reagent vocabulary (Fig. 3B). The other method of protocol initialization uses a CSV file upload similar

to the one used by the ItemType tab (Fig. 3C). This allows for bulk upload of multiple protocols, a convenient feature for adding in a large number of novel assays.

The minimum requirements for creating a new protocol include a protocol name, version number, and a protocol description. Users are encouraged to also upload a protocol file in PDF format that is stored by PEGR for users to download and print. When creating a new protocol, a user selects starting and ending materials for the protocol. These fields are within an enforced list of all ItemTypes defined in PEGR. It uses autocomplete to assist in creation. In the case of a simple buffer protocol, the individual components of the protocol are the input ItemTypes (i.e., 5M NaCl, 500 mM EDTA) and the end product is the final buffer (e.g., NaCl 250 Wash Buffer). In the case of a protocol such as PCR, the "Traced Sample" field is also used to track a sample's state entering and exiting a protocol stage. The concept of a "Traced Sample" allows PEGR to link a final sample across multiple protocols that the sample may participate in (Fig. 3D).

Similar to how ItemTypeCategory is used to organize the wide variety of ItemTypes in the "Inventory," Protocol Groups are used to consolidate and organize the variety of protocols that often compose an experiment (Fig. 3E). Protocol Groups contain any number of protocols (e.g., ChIP reaction and DNA end repair) in an ordered set. This enforces both experimental organizations and provides flexibility for a user to generate a new Protocol Group (e.g., ChIP-seq v2) by re-using previously defined protocols in combination with new protocols. ProtocolGroup's are initialized through the Admin console. This design consideration requires a ProtocolGroup to be thoroughly vetted by a PEGR administrator (i.e., Principal investigator, lab manager) before it can be accessed and used by the entire group. While users are still able to construct and initialize any individual Protocol they desire, this produces an intentional pause-point in developing novel assays which requires users (i.e., graduate students) to reflect on their experimental design and discuss with a relevant senior scientist.

### Tracking experimental metadata as it is generated

PEGR provides a section to record experiment metadata under the "Experiment" tab. In designing a section devoted to capturing experiment metadata, we considered that an experiment involves (a) inventory, (b) a protocol, (c) an input sample along with controls, and (d) a resulting product. The PEGR "Experiment" interface is designed to track and maintain the relational links between reagents (i.e., "Inventory"), protocols (i.e., "Protocol"), and the resulting end products (i.e., "Samples"). A new experiment can be initiated directly from the web interface (Fig. 4A). A new experiment can be assembled by combining any number of previously defined protocols into any desired organizational structure (Fig. 4B). Alternatively, a user can initialize a new experiment based on a Protocol Group (Fig. 4C). This provides an easy mechanism for quickly assembling common laboratory protocols and assays in a structured and well-defined manner.

Once an experiment is initialized in PEGR, the experimental metadata and status can be updated directly through the web interface. Starting a simple experiment (e.g., creating a wash buffer) allows users to add relevant inventory metadata to PEGR using a wizard-style guide that walks users through adding reagents and all their associated metadata that have previously been stored in PEGR inventory (Fig. 4D). While this can all be performed directly through the web interface, PEGR leverages a QR barcoding

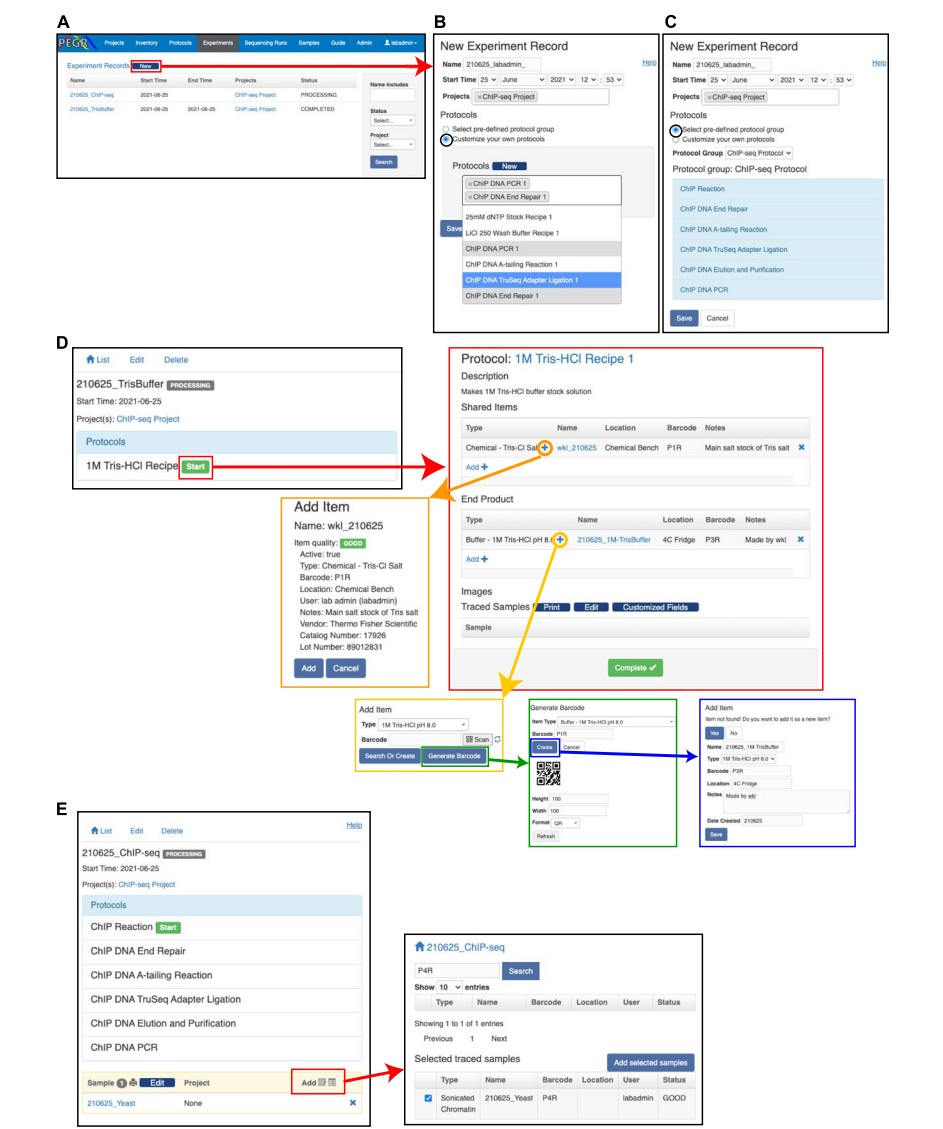

**Fig. 4** Experiment management system. **A** Browser shot of PEGR's "Experiment" interface showing all tracked experiment records. The red box and arrow display the new webform that appears when a user initializes a new experiment **B** The black circle indicates the effect of selection custom protocol construction which pulls from PEGR's approved protocol database with autocomplete. **C** The black circle indicates the effect of selecting a Protocol Group which automatically loads and manages the requirements for the entire experiment. **D** Starting the experiment (red boxes and arrow) produces a webform for users to add relevant inventory metadata to PEGR. The transition of colored boxes (orange, yellow, green, and blue) displays the workflow for adding inventory metainformation to an experiment. **E** For experiments tracking traced samples, an additional interface (red box and red arrow) appears on the webform which allows users to search PEGR's database for existing compatible reagents (i.e., cell pellet) to add to the experiment

system, similar to the inventory system, to allow users to progress through experimental stages and collect their metadata in real-time using a hand-held Android device. This information includes but is not limited to Protocol ID, Reagent ID, Equipment ID, Tech ID, date, etc. Thus, each scanned item is linked to each experiment as associated metadata. Although we display the functionality of the webform for visualization purposes,

the QR barcode scanner is the recommended method for linking experimental metadata in PEGR. In cases where the inventory item has never been previously instantiated within PEGR, a web-interface allows the user to define a new QR barcode and instance of the inventory item.

A typical lab process is to generate common laboratory reagent stocks (e.g., wash buffers) that are used multiple times across many different downstream experiments. However, more complicated experimental setups like ChIP-seq, involve a "traced" sample which moves through multiple sequential experiments and combines with different reagents as it transitions through product states (e.g., sonicated chromatin converts to DNA library). A "traced" sample typically begins as a "BioSample" in PEGR. The "BioSample" is assigned a unique "Sample" ID within the PEGR database the moment it is added to an Experiment. This provides a clear delineation in the creation of new Samples in PEGR and helps to prevent users from initializing any number of theoretical Samples that are unlinked to any Experiment. This functionality mirrors the best practices of a standard laboratory notebook. As lab notebooks are not designed to record proposed experiments, but only provides the record of a performed Experiment, this logic is consistent with standard biochemical wet-bench practices. Importantly in the case of traced samples, PEGR can display all the states that a sample has transitioned through allowing for full experimental history tracking. A traced sample can be added to an experiment using either the web-interface or the QR barcode system (Fig. 4E). Importantly, PEGR allows multiple samples to be attached to a single protocol. This enables the operator to process multiple samples in a batch while only needing to enter the related information once (e.g., when performing ChIP-seq on 8 samples in parallel).

### Sequencing and automated bioinformatic workflows

PEGR provides a section to record metadata for DNA sequencing and bioinformatic data processing. Samples processed in parallel through PEGR's "Experiment" module are natively grouped as "cohorts." Cohorts are typically generated by a researcher when addressing specific questions within a scientific project. The biochemical end for these (epi)genomic cohorts is typically high-throughput sequencing (or other detection systems) and downstream data analysis. As the throughput of DNA sequencers continues to expand, the available sequencing bandwidth for any given sequencing run will often exceed the needs of a cohort of samples. As a result, multiple cohorts are often sequenced together in a single sequencing run. These multiplexed samples may originate from different scientific projects (Fig. 5A). Reciprocally, one or more cohorts comprise a Project, to which cohorts may be added over time. Therefore, we define a "sequencing cohort" as the group of samples that belong to the sample project and a specific sequencing run.

PEGR provides substantial integration with common Illumina platforms. It implements a real-time workflow tracking and quality control dashboard through integration with external bioinformatics systems, such as the Galaxy platform (Fig. 5B) [21]. Galaxy workflows designed to communicate with PEGR contain simple XML wrappers for Python scripts which send a JSON file to PEGR RESTful API in a standard HTTP POST request. The JSON file contains a variety of information tracked by PEGR, and one critical element is the History ID from Galaxy. This allows PEGR to connect reproducible

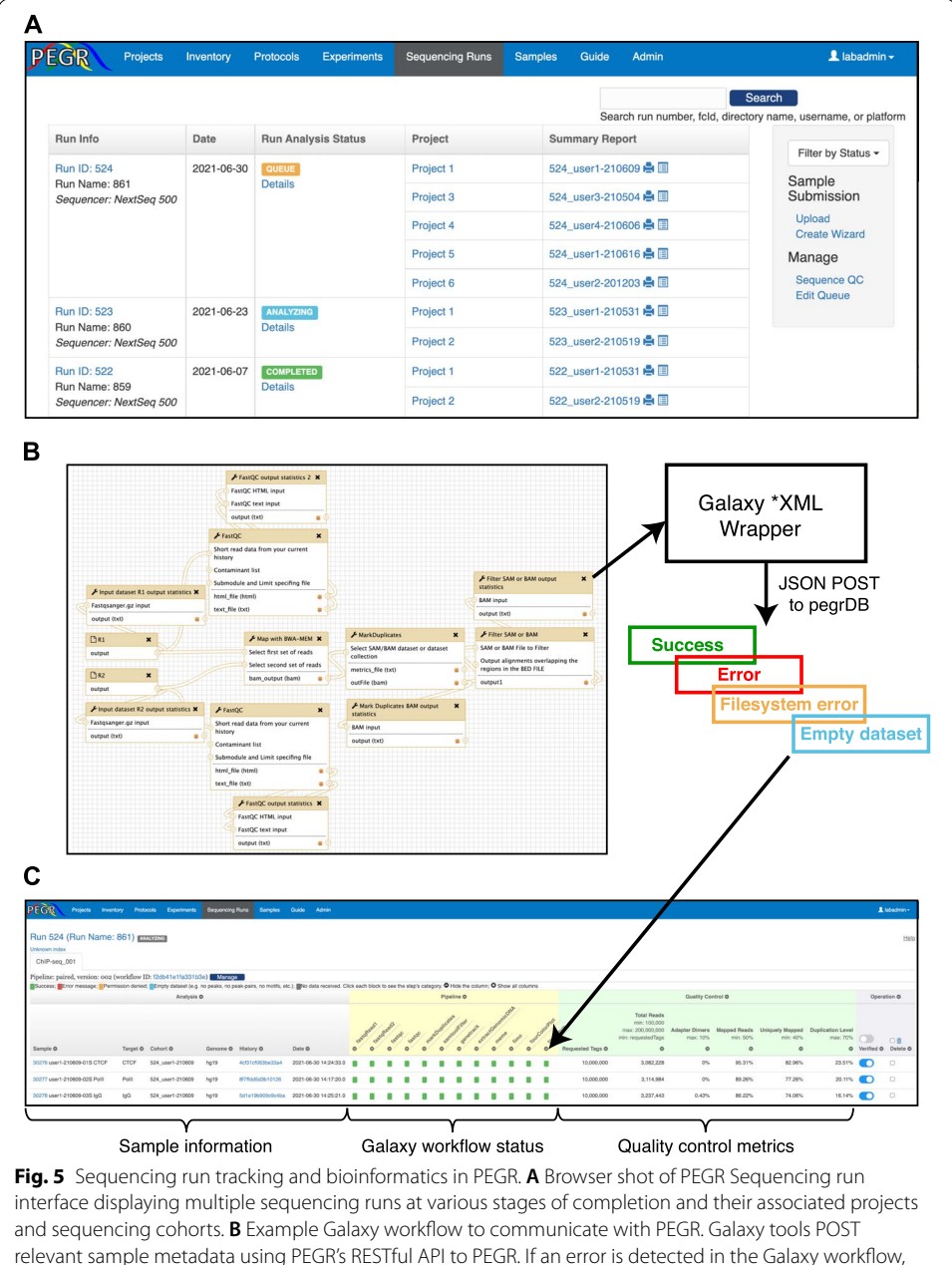

**Fig. 5** Sequencing run tracking and bioinformatics in PEGR. **A** Browser shot of PEGR Sequencing run interface displaying multiple sequencing runs at various stages of completion and their associated projects and sequencing cohorts. **B** Example Galaxy workflow to communicate with PEGR. Galaxy tools POST relevant sample metadata using PEGR's RESTful API to PEGR. If an error is detected in the Galaxy workflow, this information is also communicated to PEGR and visualized with a distinct color. **C** Browser shot of PEGR's workflow tracking and quality control dashboard. Sample information and hyperlinks to individual sample metadata is on the left, the Galaxy workflow status in the middle, and sequencing quality control statistics (e.g., read depth, mapping %) is viewable on the right

bioinformatic Galaxy workflows with the biochemical records stored and managed by PEGR [34].

As the output data from each analysis step returns to PEGR at the completion of each step, the status of the workflow is updated in real-time (Fig. 5C). The status of each analysis step being tracked is represented by a square. If the script completes successfully and passes the preliminary validation, the square will be colored in green. A script that

results in one or more error messages has its square colored red. Clicking on the square renders the error messages in detail. API calls with "permission denied" have their square colored orange, and analysis steps with missing datasets have their square colored blue. For analysis steps that have not communicated back to PEGR, the square remains gray. If all squares become green, it indicates that the entire workflow has completed successfully. Note that bioinformatic workflows may vary for different sample types, and they may include different sets of analysis steps. To accommodate different workflows, PEGR defines a configuration for each workflow that lists all the analysis steps to be tracked. The workflow tracking panel is dynamically rendered according to this configuration.

In addition to tracking the workflow-specific metadata (e.g., peak-calling completes successfully, MEME failed), PEGR also tracks assay-independent quality control metrics such as total reads per sample, adapter dimers, mapped reads, uniquely mapped reads, and PCR-duplication level (Fig. 5C). Through the web interface, the administrator may define the acceptable range for each field indicated at the header, and fields that have values outside the acceptable ranges are colored in red. This combination of statistics gives users an overview of the quality of the sequencing experiment. The thresholds for what constitutes an acceptable result are user-specified through the web-frontend of PEGR. After reviewing the statistics, Admins can indicate if the sample has passed the quality control check and been "verified." If the statistics indicates that errors may exist in the sequencing result (e.g., incorrect adapter index assignment), the authorized user can "delete" the sample directly on this page.

The workflow tracking and quality control dashboard can become quite wide as there is no upper limit to the number of scripts (i.e., columns) that may be tracked in PEGR. Users can hide columns by clicking the "−" sign on the header. The columns can be restored by clicking the "+" sign at the top. Multiple bioinformatics workflows can be applied to the samples in a single sequencing run or even a single sample. In this case, PEGR will display separate tabs for each workflow.

### Reporting, visualization, and data dissemination

The Reporting module of PEGR is an interface to report, visualize, and disseminate the data it stores. PEGR provides a "Project" interface to organize sequencing cohorts and individual samples in a reporting and visualization dashboard (Fig. 6A). The project dashboard provides links to the dynamically generated reports of entire cohorts of samples or individual samples (Fig. 6B). This interface also provides a mechanism for granting project permissions to the various users of PEGR. This allows PEGR to provide its stored sample metainformation to external collaborators while limiting their ability to access all data within PEGR that may not apply to a specific shared project.

When selecting an individual sample in PEGR, users are presented with a custom report that contains all affiliated metadata with that sample (Fig. 6C). Understanding that compliance can be difficult to achieve in certain settings, PEGR does not require a complete metadata trail for data visualization and will only display the data that it has. In addition to providing the biochemical sample metadata, PEGR will also display the results of the bioinformatic Galaxy workflows live-streamed directly from Galaxy (Fig. 6D). As a key feature, PEGR does not duplicate raw or processed data files in

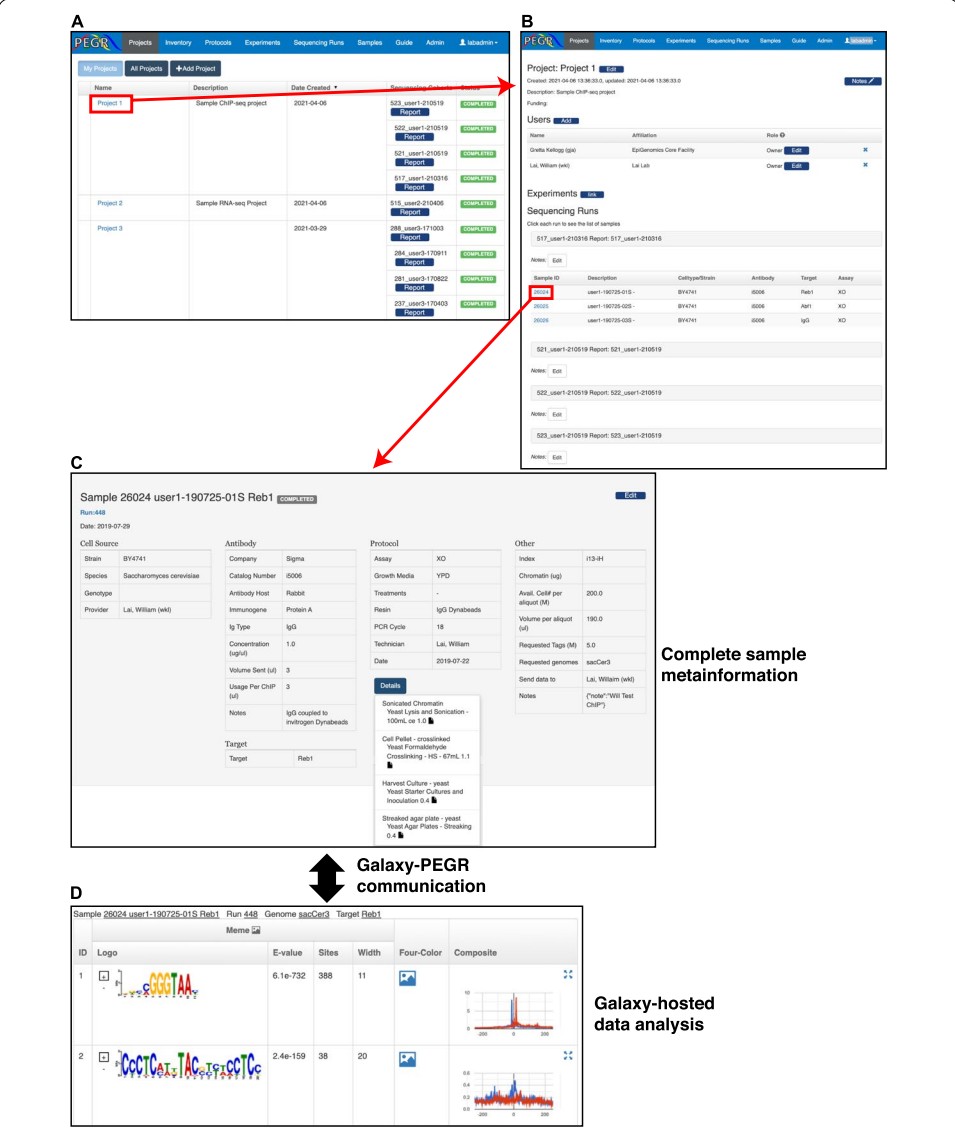

**Fig. 6** PEGR project organization and data dissemination. **A** Browser shot of the PEGR "Project" interface displaying the user's affiliated projects and the sequencing cohorts associated with each project. The red box and arrow show the transition to the individual project page. **B** Browser shot of example PEGR project displaying affiliated users, associated experiments, sequencing cohorts, and samples. The red box and arrow show the transition to the individual sample page from the affiliated project. **C** Browser shot of example sample in PEGR displaying all known metainformation and details of experiments performed with this sample. The black arrow represents the live communication with the Galaxy webserver hosting the bioinformation analyses of the sample. **D** Data hosted on Galaxy is dynamically displayed in the same PEGR web frame that reports on sample metadata to link the bioinformatics back to the experimental metadata

Galaxy. PEGR only stores the relevant metadata needed to point to sequencing datasets and downstream analysis stored on Galaxy (i.e., Galaxy HistoryID). This design choice allows PEGR to track millions of sample details with a single CPU server. The current version of PEGR used by the Cornell EpiGenomics Core tracks ~70 Tb of Galaxy-generated analyses using less than 10 Gb of hard disk space.

## Discussion

PEGR is a management platform for epigenomic and genomic research pipelines. It supports scientific data management by tracking samples from the very first step of sample preparation to the end of bioinformatics analysis and data reporting, thus supporting the FAIR principles and the reproducibility goals of the Galaxy platform [29, 35]. PEGR links people, samples, protocols, sequencers, and bioinformatics computation together, and facilitates research on genome regulation. The ability to visualize the downstream analyses of samples of interest within the same frame as the biochemical information helps users to better interpret and understand the results of their experiments.

One of the primary challenges of genomic research is how to make different experimental assays and subsequent bioinformatic analysis reproducible and available through different organizations. To tackle this challenge, we developed PEGR to provide a flexible system that maintains scientific rigor. PEGR provides a continuously tracked link from the original reagent preparation all the way through to the final downstream bioinformatic processing and figure generation. To date, PEGR has been used to support the publication of over a dozen papers comprising thousands of unique experiments across a range of distinct assays (e.g., ChIP-exo, RNA-seq, PIP-seq) [31, 32, 36–38]. PEGR is adaptable to manage and organize common genomic assays with support for tracking any metainformation deemed of value by an administrator of PEGR. Through PEGR's RESTful API, analysis results can be accepted from any client (e.g., Galaxy and Pegasus) and the workflow tracking dashboard is configurable to track any number of distinct informatic workflows. The RESTful API also provides a mechanism by which PEGR can be queried remotely to output its stored metadata to any other desired system.

## Conclusions

Future PEGR development will focus on supporting additional bioinformatic workflows and genomic assays. The current supplied bioinformatic analysis processing workflow is hard-coded to the Illumina sequencing platform. Future upgrades that can be made to PEGR include providing compatibility with non-Illumina sequencing pipelines (e.g., PacBio, Oxford Nanopore) and enhancing the sample submission process using the native web interface. Our long-term goals include enhancing role security to provide compliance with the EU GDPR, NY SHIELD, and California CCPA privacy laws for storing de-identified patient meta-information. We also believe that given the prominence of many internationally funded Galaxy instances (e.g., https://usegalaxy.org/, https://usegalaxy.eu/), a key future upgrade will be to enable multiple PEGR instances to communicate with multiple Galaxy instances in a full many-to-many relationship. This will enable researchers to directly benefit from well-funded bioinformatic rigor and reproducibility initiatives by reducing the overhead required for smaller groups to run their own Galaxy instances. These upgrades and more provide a clear path forward for providing rigorous and reproducible research across the biochemical and biomedical fields.

## Methods

### Architecture

The software stack used in the development of PEGR includes OpenJDK 11.0.12, Grails 4.0.11, and MariaDB 10.5.5. PEGR is built on Grails, a high productivity web application framework for JVM [39]. Grails follows the "coding by convention" paradigm, and provides mechanisms such as injection, templating, and scaffolding, which makes the development much more efficient. The architecture of PEGR follows the Model-View-Controller (MVC) pattern [14]. The model layer contains 75 domain classes that are mapped to the database. In support of open-source software, we chose MariaDB to host PEGR's database. The database is a relational database that contains normalized tables for various information, including users (Additional file 1: Fig. S1A), projects (Additional file 1: Fig. S1B), sample details (e.g., strain, antibody, target, and growth media) (Additional file 1: Fig. S1C), sequencing run details (Additional file 1: Fig. S1D), and bioinformatics analysis (Additional file 1: Fig. S1E). Data in the database are queried and persisted through Grails' object relational mapping (GORM). During development, the database schema often needs to be updated to meet stakeholder's requirements, e.g., adding or removing a table or a column. In this situation, we follow the code-first approach, that is, we first change the involved domain classes, run Grails database migration plugin, and the database will be automatically updated. This will guarantee synchronization between the codes and the database schema.

The controllers generate responses to clients' requests. For HTML requests, the controllers will delegate them to the views where data will be presented. Inside views, PEGR heavily utilizes JavaScript, Bootstrap library, and AJAX to improve user experience. It adopts the responsive web design and provides consistent data presentation across desktops, tablets, and mobile devices. In addition, REST-compliant (RESTful) APIs are made available that allow external applications to query data from and send data to PEGR. In response to an API request, the controllers will render the data in JSON format. Note that the business logic in the controller can become complicated. For example, it may involve nontrivial data manipulation and decision-making. In such cases, it will push the business logic to a separate layer, called "service," leaving the controllers relatively light weighted. The separation of models, controllers, views, and services confines different logic concerns to their own layer and makes it possible to reuse shared components. This enables the application to be easily developed, tested, and maintained.

### System requirements

PEGR is an extremely lightweight software platform. A production PEGR system tracking ~30,000 unique experiments across >500 sequencing runs serves multiple concurrent users with no detectable delays on a CentOS 7 server with 1 CPU, 8 Gb of RAM, and 200 Gb of hard disk space. By keeping all data analysis in a remote Galaxy webserver, data duplication and the file system footprint are kept to a minimum.

### PEGR Guide

A quick-start on installing PEGR is provided on GitHub https://github.com/seqcode/pegr, and additional tutorials on how to configure PEGR and deploy it to production are provided

on GitHub Wiki https://github.com/seqcode/pegr/wiki. Once PEGR is running, users can find the tutorial under the "Guide" tab of the PEGR interface which details the various extended capabilities of PEGR (Additional file 1: Fig. S2A).

### RESTful API

For advanced users, PEGR provides a RESTful API for users to query and download the sequencing data and analysis results. Similar to the web interface, users can query samples based on sample ID, strain, antibody, target or run ID, and PEGR will return the qualified samples along with their major statistics and the links to the sequencing datasets such as the BAM files. The multiple ways that PEGR provides for reporting and visualization make it easy to share and further utilize the sequencing data. Guidance on how to interact with PEGR and what API commands are available are detailed at "Bioinformatics API" under the PEGR Guide tab (Additional file 1: Fig. S2B). This guide provides sample code and examples on how to connect to PEGR's APIs in several common programming languages (Java, Python).

### Orchestrating bioinformatics

Since an Illumina sequencer deposits data to a pre-defined designated repository, a cron job can be set up on the PEGR-hosting server to probe the sequencer to check on the status of the new sequencing run. Once PEGR detects the completion of the sequencing run, it will match the sequencing output data with the information stored in its database and initiate a bioinformatics workflow. PEGR creates a set of files that contain the sequencing run information, e.g., the sequencing run ID, the path to the raw sequencing data repository, sample ID, the library index attached to a sample (a way to biologically identify each sample in a sequencing run), and the reference genomes to align the samples to. PEGR supports the alignment of samples to more than one reference genome (e.g., hg19 and hg38) and tracks the results of multiple alignments.

External clients, e.g., bioinformatics scripts or a workflow management tool such as Galaxy [21], can then sync with the relevant sample metainformation (i.e., genome build) and process the samples through all selected workflows. A typical processing workflow for ChIP-seq is comprised of a series of analysis steps, including raw data transformation, sequence alignment against reference genomes, peak-calling, and motif discovery. Since PEGR is charged with hosting all the metadata and final reporting, the output data needs to be communicated back to PEGR. We developed a RESTful API in PEGR to accept POST requests that contain the output data generated from the bioinformatics workflow (Fig. 5B). Each API call corresponds to a single analysis step in the workflow. When an analysis step finishes, its output data will be posted to PEGR immediately. The benefits of sending results immediately instead of gathering all the results at the end of the workflow in a batch include that (1) we can track the status of the workflow run in real-time and (2) in the event that an analysis step failed (e.g., cluster error, networking issues), we can resume the workflow from the break point.

The data sent through the API needs to be constructed in a JSON format and the fields accepted are listed in Table 1. The field "userEmail" identifies the author of the

**Table 1** PEGR API fields that accept analysis results

| Field | Format | Example |
|---|---|---|
| userEmail | String | "xxx@xxx.xxx" |
| run | Long | 123 |
| sample | Long | 10023 |
| genome | String | "sacCer3" |
| historyId | String | 99afe35e5e550d2c |
| history_url | String | https://xxx.xxx/view?id=99afe35e5e550d2c |
| alignmentId | Long | 12345 |
| workflowId | String | "b7p9d3k629e985d6" |
| workflowStepId | String | "d42c9e9785b96e77" |
| statsToolId | String | "bam_to_scidx_output_stats" |
| toolId | String | "toolshed.g2.bx.psu.edu/repos/iuc/bam_to_scidx/bam_to_scidx/1.0.1" |
| toolCategory | String | "output_bamToScidx" |
| parameters | {"parameterName": "parameterValue"} | {"dbkey":"sacCer3","require_proper_mate_pairing":0} |
| statistics | [{"statisticsName": "statisticsValue"}] | [{"genomeCoverage":0.0206}] |
| datasets | [{"type":"xxx","id":"xxx","uri":"xxx"}] | [{"type":"scidx","id":"a39ec9e14951b012","uri":"https://xxx.xxx/datasets/9e14951 a39ec b012"}] |
| toolStderr | String | "some error" |

analysis in workflow management tool and is also used to authenticate the user in combination with the API key. The fields "run" and "sample" are used to match the analysis to the sequencing run and the sample already submitted to and stored in PEGR, and "genome" is the reference genome that the sample is aligned to in this analysis. The analysis workflow and step are labeled by the corresponding "workflowId" and "workflowStepId." There could be multiple workflow runs on the same sample and each analysis step sends its results separately, therefore, we need the field "historyId" to uniquely map analysis steps to each workflow run. The software and its version used in the step are recorded in the field "toolId." In bioinformatics, many of the tools achieve the same functionality. Therefore, we also record "toolCategory" to facilitate reporting and future comparison. The "parameter" field is formatted as a dictionary with each input parameter name of the tool as the key in the dictionary and the value of the parameter as the value in the dictionary. The storage of all the parameters, along with the software and version used, helps with reproducibility and potential extension in the future. The output of the steps ranges from simple statistics (e.g., % read alignment to reference genome) to more complicated datasets (e.g., MEME motif results). The former is sent in the "statistics" field. As for the large datasets, the path to the dataset file and the file type are sent in the "datasets" field. Both "statistics" and "datasets" are formatted as a list of dictionaries.

There can be various downstream analyses following the core bioinformatics workflow. And results from those downstream analyses can be posted to PEGR through the same API and linked to its upstream alignment using the field "alignmentId." The client may also send a "note" to PEGR, such as warning or error messages.

In order to maximize flexibility and interoperability with other software platforms, PEGR is also capable of hosting links to external platforms. These tables are accessible under the Guide tab in PEGR (Additional file 1: Fig. S2B). The genome build table in the

pegrDB links to the external source of the genome (e.g., UCSC, SGD). In order to further support bioinformatic reproducibility outside of the Galaxy platform, PEGR also tracks common reference features (gene coordinates, protein binding locations, origins of replication) which can be hosted on GitHub.

### Security and privacy

PEGR leverages Spring Security to control the access to the application [40]. Spring Security is a framework that provides authentication, authorization, and other security features for enterprise applications. For authentication, PEGR uses four mechanisms: "daoAuthenticationProvider," "preAuthenticatedAuthenticationProvider," "anonymousAuthenticationProvider," and "rememberMeAuthenticationProvider." Through the "daoAuthenticationProvider," users can log in to PEGR using their registered username and password. The "preAuthenticatedAuthenticationProvider" works with external identity hosts so that users do not have to create an additional password with PEGR. It supports both header-based and attribute-based authentication, e.g., CoSign (http://weblogin.org) and Shibboleth (https://www.shibboleth.net) Single Sign-on. The "anonymousAuthenticationProvider" denies access to the application from unauthorized users except for the login page. and the "rememberMeAuthenticationProvider" stores the login information in a cookie and allows for automatic login for an extended period.

PEGR authorizes data access through a number of security layers. Primarily, PEGR implements role-based access control which is mostly implemented through URL intercept. It assigns a role group to each user, and users in a specific role group can only access the URLs authorized for that role group. For example, a user who is assigned to the "Admin" role group will have access to all the URLs, including those for the Admin console, and be able to update all the samples and projects. In contrast, users in the "Member" role group cannot access certain URLs, e.g., those for the Admin console. "Members" do have "read" access to all projects where they have been assigned, including the "Inventory," "Lab Protocols," "Experiments," "Sequencing Runs," and "Samples." Members do not have "write" access to those objects unless they have a specific "project role" (see below) or ownership to an inventory item or protocol. A user cannot access any part of PEGR until their ID is assigned to a role group. By default, users are initially added to the "Guest" role group with the most restricted access in the PEGR platform, only able to view unprotected information.

Additionally, more granular access controls are defined for each project and sample through controller interceptors. The controller interceptors apply certain logics across a group of controller actions, and they are executed before the related controller actions are invoked. For example, users can be added to a project with different roles, (e.g., "owner", "participant", or "guest"). While all users linked to the project can view the project summary and all the samples in the project, only the owners and participants of the project have the ability to add (or remove) samples to (or from) the project and edit the samples that belong to the sample. And only the owners of the project can edit the project's information, such as project name, project description, and funding source.

## Supplementary Information

---

**Additional file 1.** Integrated supplementary figures S1 and S2.

**Additional file 2.** Review history.

---

### Acknowledgements

The authors thank Greg Von Kuster, Bongsoo Park, Geoffrey Billy, Belinda Giardine, Abeer Almutairy, Hedgie Jo, and Pierce Chafflin for their user testing and feedback, code base tests, development code contributions, and helpful discussions and feedback to this project. Computations for this research were performed on the Pennsylvania State University's Institute for Computational and Sciences' ROAR supercomputer (Galaxy, RRID:SCR_006281).

### Peer review information

### Review history

The review history is available as Additional file 2.

### Authors' contributions

D.S. is the software engineer who built the computational platform. IT infrastructure architecture and database schema design were provided by G.D.K., S.M., and W.K.M.L.. B.F.P., S.M., and W.K.M.L. provided design specifications and biochemical expertise. A.N. and P.K. architected the current Galaxy workflow communication system. D.S. and W.K.M.L. co-wrote the manuscript. The authors read and approved the final manuscript.

### Authors' information

Twitter handles: @GrettaKellogg (Gretta D Kellogg); @prashkuntala (Prashant K Kuntala); @mahonylab (Shaun Mahony); @ThePughLab (B. Franklin Pugh); @WilliamKMLai (William KM Lai).

### Funding

This work was supported by the US National Institutes of Health (NIH) grants 5R01-ES013768 and 3R01-GM125722-03S1 for funding the development and dissemination of PEGR. The authors acknowledge the support of the Institute for Computational and Data Sciences at the Pennsylvania State University through ICDS Seed Grants to Dr. B. Franklin Pugh and Dr. William KM Lai.

### Availability of data and materials

The PEGR software is publicly available at https://github.com/seqcode/pegr [41]. A Docker image of PEGR is also available on Docker Hub: https://hub.docker.com/repository/docker/dshao/pegr.The PEGR-Galaxy communication scripts are available at https://github.com/CEGRcode/pegr-galaxy_tools and the PEGR NGS pipeline scripts are available at https://github.com/CEGRcode/pegr-ngs_pipeline under the MIT license. A release of the PEGR source code (v0.3.1) is available on Zenodo: https://doi.org/10.5281/zenodo.6401788 [42]. When PEGR software is used as the platform providing the data, it can be cited using the RRID:SCR_021861.

## Declarations

### Ethics approval and consent to participate

Not applicable.

### Consent for publication

Not applicable.

### Competing interests

The authors declare that they have no competing interests.

### Author details

[1]Institute for Computational and Data Sciences, Pennsylvania State University, University Park, PA 16802, USA. [2]Cornell Institute of Biotechnology, Cornell University, Ithaca, NY 14850, USA. [3]Department of Biochemistry & Molecular Biology, Pennsylvania State University, University Park, PA 16802, USA. [4]Department of Molecular Biology and Genetics, Cornell University, Ithaca, NY 14850, USA. [5]Department of Computational Biology, Cornell University, Ithaca, NY 14850, USA.

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

## 