## [**Additional file 2.** Review history. · Genome Biology]

Review History

First round of review

Reviewer 1

Were you able to assess all statistics in the manuscript, including the appropriateness of statistical tests used? There are no statistics in the manuscript.

Were you able to directly test the methods? Yes.

Comments to author:

Software Testing:

Starting with a fresh Ubuntu 20.04.3 virtual machine running under VirtualBox that was updated and had required packages installed, I was able to successfully compile PEGR from source, resulting in a "pegr-0.1.war" that I was able to successfully run and interact with. However, for the rest of this review I used the latest officially released pre-compiled pegr.war file provided (<https://github.com/seqcode/pegr/releases/tag/v0.2.6>).

Barcode Scanner to add samples to experiment does not work. Clicking the QR/Barcode link will launch the android app, and then scanning barcodes will work, and then automatically load back into the PEGR website, but the textbox is not filled in with the barcode value.

"Samples" will not be populated until a BioSample that has been added to inventory is added to an experiment.

When creating BioSample inventory (sonicated chromatin) and selecting Genus: "Homo", Species: "Sapien", and clicking save, it incorrectly saves the values as *Saccharomyces cerevisiae*. Going into edit, and changing to *Homo sapien*, it then requires setting "strain" to a value in order to save. I did create several *S. cerevisiae* samples first, and they can be created without strain, but if you go into edit and then try to save, you are forced to fill out strain to save these as well. It is unclear why strain is always required.

I did not have an illumina sequencer available to fully test instrument support, but I created a sequence run in PEGR UI. Clicking the "Add Master Pool" results in an empty screen in web browser, and a traceback in shell log. Errors (and ideally suggestions to fix) should be displayed to the user, or at least an indication of an error instead of an empty page.

When adding inventory to protocol instance, if you are creating new, and you do not set a barcode under "search", it will allow you to save and go to the next stage and let you name the new item, but you cannot set barcode value, and trying to save results in an error printed only to

shell logs.

When trying to add inventory item to protocol instance, if you enter a barcode for the wrong itemtype, you get a blank screen in the browser, but no error message -- there is an error in the shell log.

Two example workflows, listed under "Pipelines" in PEGR cannot be accessed, they are behind a PSU.edu forced login:

https://chipexo-gw.aci.ics.psu.edu/workflow/display_by_id?id=f2db41e1fa331b3e

https://chipexo-gw.aci.ics.psu.edu/workflow/display_by_id?id=a799d38679e985db

Also, it seems that the above PSU.edu Galaxy instance has been configured without a proper secret key, as the encoded workflow ids "f2db41e1fa331b3e" and "a799d38679e985db" matches integer 1 and 6, using the default non-secure "id_secret". Recommend following instructions from Galaxy project on setting up a secure production server.

While LIMS systems are of interest, it is the integration of PEGR with Galaxy and other potential Workflow/Analysis systems that is the novel (and dare I say "interesting") contribution. Briefly, the interconnection between PEGR and Galaxy utilizes a set of scripts run on the computer where the "NGS repo" is located (e.g. where sequencing datasets are deposited by illumina sequencer) that will upload data to Galaxy and then execute workflows. For Galaxy to work with PEGR, it must have a custom set of tools installed that are able to report back status and statistics about their parent jobs. These custom tools then POST JSON values to the PEGR server. PEGR provides its own API to receive these posts from the custom Galaxy Tools. Additionally, the workflow being called in Galaxy must have parallel Steps defined inside of PEGR, which will be used to relay status.

This design seems less than ideal. First, it requires the manual installation of non-standard tools into a Galaxy instance -- tools are not installable from ToolShed. There is one new configuration file per Galaxy instance that provides the details for these custom tools to interact with PEGR API. This essentially causes the case where you need to have a Custom Galaxy instance that is specific to a particular PEGR server, and vice versa. A single Galaxy server cannot work with more than 1 PEGR server and a PEGR server cannot work with more than 1 Galaxy server. And a PEGR instance must be specifically configured with a specific Galaxy server and the Galaxy server must be specifically configured to work with that PEGR server. There is no way to e.g. take advantage of a general use institutional Galaxy server. Having to duplicately define workflow steps in PEGR and Galaxy seems unnecessary, PEGR could parse Galaxy API description of Workflow.

An example of one of these custom tools is "BWA-MEM single read output statistics", which takes a BAM file as input, provides a python file that e.g. calls "samtools view" and may perform some computations in e.g. numpy and then builds up a custom PEGR-only JSON file containing statistics and POSTs back to PEGR. There doesn't seem to be any concern for e.g. a standard

Galaxy User coming along and running the custom tools and potentially inadvertently adding information to the pegrDB.

A more generalized approach would seem to be to use/design standard tools that can produce the statistics needed, which could be available from the ToolShed and could be useful to standard Galaxy users; then have PEGR interact directly with the Galaxy API. The two potential negatives that this would introduce would be that it would require PEGR to poll Galaxy for job completion (instead of just waiting for a POST) and some minimal parsing of e.g. text files with tables containing values, instead of JSON. But this approach would allow e.g. any number of PEGR servers to work with a single Galaxy server. And in fact, could be configured to allow a single PEGR server to work with multiple Galaxy servers for e.g. different workflows or organisms, etc. Could the authors please comment on these points?

The instructions on configuring PEGR with analysis workflow/pipeline support is beyond minimal, but is available by inspecting these github repos: "The PEGR-Galaxy communication scripts are available at https://github.com/CEGRcode/pegr-galaxy_tools and the PEGR NGS pipeline scripts are available at https://github.com/CEGRcode/pegr-ngs_pipeline under the MIT license."

There must be clear instructions for installing and enabling the advertised interconnectivity of PEGR and Galaxy. More reliance on Galaxy API could simplify a lot of the configuration currently needed within PEGR.

The PEGR-Galaxy communication scripts are available at https://github.com/CEGRcode/pegr-galaxy_tools

Listed in this manuscript (created 3 months ago):

```
$ git clone https://github.com/CEGRcode/pegr-galaxy_tools.git
Cloning into 'pegr-galaxy_tools'...
remote: Enumerating objects: 65, done.
remote: Counting objects: 100% (65/65), done.
remote: Compressing objects: 100% (24/24), done.
remote: Total 65 (delta 37), reused 65 (delta 37), pack-reused 0
Unpacking objects: 100% (65/65), 121.55 KiB | 313.00 KiB/s, done.
```

An unmentioned repository created 5 years ago, with additional Primary contributors not listed as authors nor listed under acknowledgements of current manuscript:

```
$ git clone https://github.com/seqcode/cegr-galaxy.git
Cloning into 'cegr-galaxy'...
remote: Enumerating objects: 1570, done.
remote: Counting objects: 100% (72/72), done.
```

```
remote: Compressing objects: 100% (22/22), done.
remote: Total 1570 (delta 49), reused 59 (delta 43), pack-reused 1498
Receiving objects: 100% (1570/1570), 1.65 MiB | 337.00 KiB/s, done.
Resolving deltas: 100% (928/928), done.
```

There are no differences between the tools in these separate repositories:
\$ diff -r cegr-galaxy/tools/cegr_statistics/ pegr-galaxy_tools/tools/

It is not clear why a new repo was created 3 months ago with the tools manually copied over from a 5 year old repository without including attribution of the primary committer.

the PEGR NGS pipeline scripts are available at https://github.com/CEGRcode/pegr-ngs_pipeline

```
Listed in this manuscript (created 3 months ago):
$ git clone https://github.com/CEGRcode/pegr-ngs_pipeline.git
Cloning into 'pegr-ngs_pipeline'...
remote: Enumerating objects: 25, done.
remote: Counting objects: 100% (25/25), done.
remote: Compressing objects: 100% (24/24), done.
remote: Total 25 (delta 0), reused 25 (delta 0), pack-reused 0
Unpacking objects: 100% (25/25), 29.46 KiB | 1.40 MiB/s, done.
```

There is no difference between these "xEGR NGS pipeline scripts" (5 years ago, 3 months ago):
\$ diff -r cegr-galaxy/scripts/ pegr-ngs_pipeline/scripts/

There are minimal differences between "cegr" and "pegr" sample configurations, such as changing naming and removing references to a home directory of user "gvk":

```
$ diff cegr-galaxy/config/cegr_config.ini.sample pegr-
ngs_pipeline/config/pegr_config.ini.sample
8,9c8,9
< ANALYSIS_PREP_LOG_FILE_DIR = /Users/gvk/work/git_workspace/cegr_galaxy/log
< API_KEY = 948f9b9e37c1ff5d39d6ea1a6cf13e46
---
> ANALYSIS_PREP_LOG_FILE_DIR = /pegr_galaxy/log
> API_KEY = XXXXXXXXXXXXXXXXXXXXXXXXXXXXXXXXXXXXXXXX
11,12c11,12
< BCL2FASTQ_BINARY = /Users/gvk/work/bcl2fastq_binary
< BCL2FASTQ_REPORT_DIR = /Users/gvk/work/bcl2fastq_reports
---
> BCL2FASTQ_BINARY = /path/to/bin/bcl2fastq_binary
> BCL2FASTQ_REPORT_DIR = /path/to/bin/bcl2fastq_reports
```

```
18c18
< # this enhancement was added manually to the ~/src/FastQValidator.cpp file:
---
> # this enhancement was added manually to the ~/src/FastQValidator.cpp file:
21c21
< FASTQ_VALIDATOR_BINARY = /Users/gvk/work/fastQValidator
---
> FASTQ_VALIDATOR_BINARY = /path/to/bin/fastQValidator
24c24
< GALAXY_HOME = /Users/gvk/work/git_workspace/galaxy
---
> GALAXY_HOME = /galaxy
26c26
< LIBRARY_PREP_DIR = /Users/gvk/work/git_workspace/cegr_galaxy/config/library_prep_dir
---
> LIBRARY_PREP_DIR = /pegr_galaxy/config/library_prep_dir
29c29
< PREP_VIRTUAL_ENV =
/Users/gvk/work/git_workspace/cegr_galaxy/venv/bin/activate_this.py
---
> PREP_VIRTUAL_ENV = /pegr_galaxy/venv/bin/activate_this.py
31,32c31,32
< RAW_DATA_DIR = /Users/gvk/work/git_workspace/raw_data_dir
< RAW_DATA_LOGIN = nextseq@146.186.153.198
---
> RAW_DATA_DIR = /raw_data_dir
> RAW_DATA_LOGIN = nextseq@123.456.78.90
35c35
< REMOTE_RUN_INFO_FILE = /home/nextseq/NSQData_PughLab/cegr_run_info.txt
---
> REMOTE_RUN_INFO_FILE = /home/nextseq/NSQData_PughLab/pegr_run_info.txt
37,38c37,38
< REMOTE_WORKFLOW_CONFIG_DIR_NAME = cegr_config
< RUN_INFO_FILE = /Users/gvk/work/git_workspace/cegr_galaxy/config/cegr_run_info.txt
---
> REMOTE_WORKFLOW_CONFIG_DIR_NAME = pegr_config
> RUN_INFO_FILE = /pegr_galaxy/config/pegr_run_info.txt
40c40
< SAMPLE_SHEET =
/Users/gvk/work/git_workspace/cegr_galaxy/config/cegr_sample_sheet.csv
---
> SAMPLE_SHEET = /pegr_galaxy/config/cegr_sample_sheet.csv
63,64c63,64
< SINGLE = single_001
< PAIRED = paired_001
---
```

```
> SINGLE = single_002
> PAIRED = paired_002
82,84d81
< NC003552 = /nfs/brubeck.bx.psu.edu/scratch5/galaxy-pugh/galaxy/tool-
data/NC003552/len/NC003552.len
< pa01 = /nfs/brubeck.bx.psu.edu/scratch5/galaxy-pugh/galaxy/tool-data/pa01/len/pa01.len
< pf25 = /nfs/brubeck.bx.psu.edu/scratch5/galaxy-pugh/galaxy/tool-data/pf25/len/pf25.len
86,88d82
< sp8 = /nfs/brubeck.bx.psu.edu/scratch5/galaxy-pugh/galaxy/tool-data/sp8/len/sp8.len
< # TODO: tair10.len is not available in this Galaxy instance
< # tair10 = /nfs/brubeck.bx.psu.edu/scratch5/galaxy-pugh/galaxy/tool-data/
```

It is not clear why a new repo was created 3 months ago with the scripts manually copied over from a 5 year old repository without including attribution of the primary committer.

It would be helpful if the authors could highlight the specific differences between "PEGR" and "CEGR" [my apologies if this is not the correct naming for this previously existing system], in particular the interconnectivity of the LIMS and analysis workflow systems (e.g. Galaxy).

Reviewer 2

Were you able to assess all statistics in the manuscript, including the appropriateness of statistical tests used? There are no statistics in the manuscript.

Were you able to directly test the methods? No.

Comments to author:

The paper introduces a web-based platform for sample tracking and project management of genomic experiments that integrates with bioinformatics analysis performed on Galaxy. The paper is well written and provides sound justification for the approach and good technical details, with several screenshots and diagrams that clarify how the system works.

An initial version of the platform was presented in 2020 on ACM Intl. Conference Proceeding Series, and although it is still probably early days, it would be nice if the authors could include some numbers on user uptake, or at least a description of its impact where it was deployed at their labs.

I have tried without success to install PEGR following the instruction on GitHub (see suggestion below about providing a Docker image or demo server), so I'll have to base my comments on the description given in the paper.

Integrating platforms like LIMS and workflow management systems is a complex undertaking (especially for an open source project), where there is a risk of creating a platform that is too complex or not good enough (in terms of features or usability) for the single tasks. In this respect, I support the authors' approach of not reinventing the wheel for the analysis subsystem, although the way in which Galaxy tools are integrated seems half-baked (see detailed comment

below).

In conclusion, the paper and the software address the important issue of extending reproducibility from the wet lab up to the final report of a bioinformatic analysis, and I hope the authors succeed in popularising PEGR and continue its development.

Comments:

- Background, 1st paragraph: references 4-11 seem very general and not particularly relevant to the reproducibility issues described.
- Page 5, 1st paragraph: "To prevent disorganization ..., ItemType's are organized into an ItemTypeCategory" - maybe replace "organized" with "grouped". Also, is this a proper hierarchical structure (with ItemCategory's grouped in a higher ItemCatoregory)?
- Page 5: Does PEGR comw with some predefined ItemType's? Would a AppStore-like website be helpful for sharing definitions among PEGR admins?
- Figure 3 caption: "(E) 'ProtocolGroup' is accessible through the Admin console" - Why can ProtocolGroup's can be created only by admins? This seems something that can become a bottleneck.
- Page 9, 3rd paragraph: "A traced sample can be added to an experiment using either the web-interface or the QR barcode system" - Can samples be added in advance (before creating a new Experiment? There is a "Samples" button in the top menu.
- Page 24, Availability of data and materials: I'd suggest to also make PEGR available as a Docker image and/or a demo server. I've tried to run it following the instructions at <https://github.com/seqcode/pegr> but ended with a long page full of Java exceptions.
- Page 20: "When an analysis step finishes, its output data will be posted to PEGR immediately." - It's not clear how this happens, was Galaxy modified to initiate a HTTP POST request towards the PEGR RESTful API after a tool is run, or is it the tool's responsibility? By looking at https://github.com/CEGRcode/pegr-galaxy_tools it seems it is the latter, in fact each Galaxy tool published there sends its output to PEGR autonomously. If that is the case, this is a serious limitation to the usefulness of PEGR, because users wouldn't be able to directly make use of the thousands of Galaxy tools available on the Galaxy ToolShed.

Minor comments:

- Background, 1st paragraph: "What is needed is systematic metadata capture..." - This sentence could be made a bit more absolute, something like "One way to tackle these issues is to apply systematic metadata capture..."
- Background, 3rd paragraph: <https://doi.org/10.1093/bioinformatics/btt115> is an (abandoned) attempt to extend Galaxy with LIMS functionalities that may be worth to mention.
- Background, 4th paragraph: "Galaxy.org" -> "Galaxy"
- Page 7, last paragraph: "organize the variety of protocols that often come sequentially in a pipeline": do protocols have to come in a sequence? I think it should be possible to execute some steps in parallel, so maybe "organize the variety of protocols that often compose an experiment"
- Page 9, 1st paragraph: "The PEGR 'Experiment' interface is designed to track and maintain the relational links between reagents, protocols, and the resulting end products" - I think that "samples" should be listed here even if they are formally introduced some lines below.
- Figure 4 caption: "...the effect of electing a Protocol Group" - "electing" -> "selecting"
- Page 11, 2nd paragraph: "...XML-wrapper python scripts which send a JSON file to PEGR in a standard POST request." -> "...XML-wrappers for Python scripts which send a JSON file to

PEGR RESTful API in a standard HTTP POST request."

- Figure 6 caption: "...the users affiliated projects..." -> "...the user's affiliated projects..."

- Page 16, 1st paragraph: "It supports reproducibility... management [22, 26]" - A bit unreadable, I would rewrite as "It supports scientific data management by tracking samples from the very first step of sample preparation to the end of bioinformatics analysis and data reporting thus supporting the FAIR principles [22] and the reproducibility goals of the Galaxy platform [26]."

- Supplemental Figure 1: It would be useful to include or link the source file used to generate this figure, the diagrams are too small to be useful/readable.

We kindly thank the reviewers for their critical assessments. Our responses to the issues raised are **written in bold**. Pertinent modified text in the manuscript is *italic small text*.

Reviewer #1: Software Testing:

Starting with a fresh Ubuntu 20.04.3 virtual machine running under VirtualBox that was updated and had required packages installed, I was able to successfully compile PEGR from source, resulting in a "pegr-0.1.war" that I was able to successfully run and interact with. However, for the rest of this review I used the latest officially released pre-compiled pegr.war file provided (<https://github.com/seqcode/pegr/releases/tag/v0.2.6>).

1. Barcode Scanner to add samples to experiment does not work. Clicking the QR/Barcode link will launch the android app, and then scanning barcodes will work, and then automatically load back into the PEGR website, but the textbox is not filled in with the barcode value.

We are extremely grateful to the reviewer for the extensive list of detected issues and have attempted to address as many of them as possible which we detail below. In attempting to reproduce this bug, we used the 'Barcode Scanner' app by 'ZXing' on an Android device as described in the manuscript and were unable to reproduce this issue. While we were unable to reproduce this particular bug, we note that we have now upgraded PEGR to run on Grails 4 and Java 11. We are hopeful that between this large-scale version upgrade and the several dozen bug fixes we have implemented since our initial submission, that this particular issue has been resolved.

The software stack used in the development of PEGR includes OpenJDK 11.0.12, Grails 4.0.11 and MariaDB 10.5.5. PEGR is built on Grails, a high productivity web application framework for JVM [39].

2. "Samples" will not be populated until a BioSample that has been added to inventory is added to an experiment.

We agree with the reviewer that this is a potentially unclear aspect of PEGR's design. This design consideration was done in strong collaboration with Dr. Frank Pugh, leveraging his extensive experience in biochemical experiment design and implementation. The question of when a sample becomes a 'Sample' was determined to be the point at which it becomes involved within an experiment. The utility of initializing a Sample before an experiment was performed seemed problematic as this could potentially encourage users to initialize any number of 'theoretical' Samples within PEGR that are not linked to any actually performed experiment. We have further expanded the text within the Experiment section to better explain the rationale for why this is an important distinction in biochemical experimental design.

A typical lab process is to generate common laboratory reagent stocks (e.g., wash buffers) that are used multiple times across many different downstream experiments. However, more complicated experimental setups like ChIP-seq, involve a 'traced' sample which moves through multiple sequential experiments and combines with different reagents as it transitions through product states (e.g., sonicated chromatin converts to DNA library). A 'traced' sample typically begins as a 'BioSample' in PEGR. The 'BioSample' is assigned a unique 'Sample' ID within the PEGR database the moment it is added to an Experiment. This provides a clear delineation in the creation of new Samples in PEGR and helps to prevent users from initializing any number of theoretical Samples that are unlinked to any Experiment. This functionality mirrors the best practices of a standard laboratory notebook. As lab notebooks are not designed to record proposed experiments, but only provides the record of a performed Experiment, this logic is consistent with standard biochemical wet-bench practices. Importantly in the case of traced samples, PEGR can display all the states that a sample has transitioned through allowing for full experimental history tracking. A traced sample can be added to an experiment using either the web-interface or the QR barcode system (Figure 4E). Importantly, PEGR allows multiple samples to be attached to a single protocol. This enables the operator to process multiple samples in

a batch while only needing to enter the related information once (e.g., when performing ChIP-seq on 8 samples in parallel).

3. When creating BioSample inventory (sonicated chromatin) and selecting Genus: "Homo", Species: "Sapien", and clicking save, it incorrectly saves the values as *Saccharomyces cerevisiae*. Going into edit, and changing to *Homo sapien*, it then requires setting "strain" to a value in order to save. I did create several *S. cerevisiae* samples first, and they can be created without strain, but if you go into edit and then try to save, you are forced to fill out strain to save these as well. It is unclear why strain is always required.

We were unable to reproduce the error as described by the reviewer. However as noted above, we hope that the significant upgrades performed since submission have ameliorated this particular issue. We also agree with the reviewer that 'Strain' should not be a required field for initializing a BioSample and we have removed that requirement from PEGR:

(<https://github.com/seqcode/pegr/commit/96fecc3ff6b94f14c5174d04bd6f5b1198368c57>)

4. I did not have an illumina sequencer available to fully test instrument support, but I created a sequence run in PEGR UI. Clicking the "Add Master Pool" results in an empty screen in web browser, and a traceback in shell log. Errors (and ideally suggestions to fix) should be displayed to the user, or at least an indication of an error instead of an empty page.

We were also unable to reproduce the error as described by the reviewer, however many of the bug fixes made in the repo since submission directly address outputting StdErr to the user on the web-front end instead of solely through logging files a standard user may not have access to.

Example updates:

(<https://github.com/seqcode/pegr/commit/fabc6c01177a93cd33c54d08efdb7e6b5f78d98d>)

(<https://github.com/seqcode/pegr/commit/4406fc96c6bc67b21f876fcb79de5b1721bcd6e9>)

(<https://github.com/seqcode/pegr/commit/bacacb9bd4cac3bddcb66286f15f3975e95c26cd>)

(<https://github.com/seqcode/pegr/commit/4fca2e4175a75299a0cc3e9105bfb7d2c2039f68>)

5. When adding inventory to protocol instance, if you are creating new, and you do not set a barcode under "search", it will allow you to save and go to the next stage and let you name the new item, but you cannot set barcode value, and trying to save results in an error printed only to shell logs.

We thank the reviewer for identifying this issue. It should now be resolved in the latest version of PEGR:

(<https://github.com/seqcode/pegr/commit/bad84a21393a9c2f45debd998766417fe02cd901>)

6. When trying to add inventory item to protocol instance, if you enter a barcode for the wrong itemtype, you get a blank screen in the browser, but no error message -- there is an error in the shell log.

We thank the reviewer for identifying the lack of error messaging to the web front end user. We have update PEGR to appropriately inform the user of the error:

(<https://github.com/seqcode/pegr/commit/4fca2e4175a75299a0cc3e9105bfb7d2c2039f68>)

7. Two example workflows, listed under "Pipelines" in PEGR cannot be accessed, they are behind a PSU.edu forced login:

https://chipexo-gw.aci.ics.psu.edu/workflow/display_by_id?id=f2db41e1fa331b3e

https://chipexo-gw.aci.ics.psu.edu/workflow/display_by_id?id=a799d38679e985db

Also, it seems that the above PSU.edu Galaxy instance has been configured without a proper secret key, as the encoded workflow ids "f2db41e1fa331b3e" and "a799d38679e985db" matches integer 1 and 6, using the default non-secure "id_secret". Recommend following instructions from Galaxy project on setting up a secure production server.

We apologize for the error in workflow availability. As the reviewer noted, the default PEGR workflows were incorrectly pointing to a private Galaxy development instance. We have now updated the baseline SQL database to appropriately point at publicly available workflows:

(<https://github.com/seqcode/pegr/commit/edb7bba3b1a28cb47b7701793150ef07298a91a3>)

(https://raw.githubusercontent.com/CEGRcode/pegr-galaxy_tools/main/workflows/paired_002.ga)

(https://raw.githubusercontent.com/CEGRcode/pegr-galaxy_tools/main/workflows/single_002.ga)

8. While LIMS systems are of interest, it is the integration of PEGR with Galaxy and other potential Workflow/Analysis systems that is the novel (and dare I say "interesting") contribution.

Briefly, the interconnection between PEGR and Galaxy utilizes a set of scripts run on the computer where the "NGS repo" is located (e.g. where sequencing datasets are deposited by illumina sequencer) that will upload data to Galaxy and then execute workflows. For Galaxy to work with PEGR, it must have a custom set of tools installed that are able to report back status and statistics about their parent jobs. These custom tools then POST JSON values to the PEGR server. PEGR provides its own API to receive these posts from the custom Galaxy Tools. Additionally, the workflow being called in Galaxy must have parallel Steps defined inside of PEGR, which will be used to relay status.

This design seems less than ideal. First, it requires the manual installation of non-standard tools into a Galaxy instance -- tools are not installable from ToolShed. There is one new configuration file per Galaxy instance that provides the details for these custom tools to interact with PEGR API. This essentially causes the case where you need to have a Custom Galaxy instance that is specific to a particular PEGR server, and vice versa. A single Galaxy server cannot work with more than 1 PEGR server and a PEGR server cannot work with more than 1 Galaxy server. And a PEGR instance must be specifically configured with a specific Galaxy server and the Galaxy server must be specifically configured to work with that PEGR server. There is no way to e.g. take advantage of a general use institutional Galaxy server. Having to duplicately define workflow steps in PEGR and Galaxy seems unnecessary, PEGR could parse Galaxy API description of Workflow.

An example of one of these custom tools is "BWA-MEM single read output statistics", which takes a BAM file as input, provides a python file that e.g. calls "samtools view" and may perform some computations in e.g. numpy and then builds up a custom PEGR-only JSON file containing statistics and POSTs back to PEGR. There doesn't seem to be any concern for e.g a standard Galaxy User coming along and running the custom tools and potentially inadvertently adding information to the pegrDB.

A more generalized approach would seem to be to use/design standard tools that can produce the statistics needed, which could be available from the ToolShed and could be useful to standard Galaxy users; then have PEGR interact directly with the Galaxy API. The two potential negatives that this would introduce would be that it would require

PEGR to poll Galaxy for job completion (instead of just waiting for a POST) and some minimal parsing of e.g. text files with tables containing values, instead of JSON. But this approach would allow e.g. any number of PEGR servers to work with a single Galaxy server. And in fact, could be configured to allow a single PEGR server to work with multiple Galaxy servers for e.g. different workflows or organisms, etc. Could the authors please comment on these points?

We agree with the reviewer that the design for PEGR-Galaxy communication is not currently optimized and is a point that was also raised by Reviewer 2. As this reviewer noted, the dependency on Galaxy sending communication to PEGR provides the significant upside of not requiring PEGR to continually poll Galaxy for job status. This functionality would have essentially unnecessarily reproduced the workflow functionality of Galaxy. The downside of this design is that while PEGR can communicate with multiple Galaxy instances, a true many-to-many relationship between multiple Galaxy and PEGR instances is not currently supported. We have now added text to the Discussion section describing the future of PEGR-Galaxy tool development and how we plan to allow for a true many-to-many relationship between PEGR and Galaxy instances.

Future PEGR development will focus on supporting additional bioinformatic workflows and genomic assays. The currently supplied bioinformatic analysis processing workflow is hard-coded to the Illumina sequencing platform. Future upgrades that can be made to PEGR include providing compatibility with non-Illumina sequencing pipelines (e.g., PacBio, Oxford Nanopore) and enhancing the sample submission process using the native web interface. Our long-term goals include enhancing role security to provide compliance with the EU GDPR, NY SHIELD, and California CCPA privacy laws for storing de-identified patient meta-information. We also believe that given the prominence of many internationally funded Galaxy instances (e.g., <https://usegalaxy.org/>, <https://usegalaxy.eu/>), a key future upgrade will be to enable multiple PEGR instances to communicate with multiple Galaxy instances in a full many-to-many relationship. This will enable researchers to directly benefit from well-funded bioinformatic rigor and reproducibility initiatives by reducing the overhead required for smaller groups to run their own Galaxy instances. These upgrades and more provide a clear path forward for providing rigorous and reproducible research across the biochemical and biomedical fields.

We are also keenly aware that increased generalization of the PEGR-Galaxy communication tools would be beneficial and supportive of greater community adoption, and we have now taken steps to produce a general PEGR-Galaxy communication tool that can send a status update to PEGR from any given Galaxy tool:

(https://github.com/CEGRcode/pegr-galaxy_tools/commit/06f836cf2fb46e452119ad8f6c45650cb6a2e370)

9. The instructions on configuring PEGR with analysis workflow/pipeline support is beyond minimal, but is available by inspecting these github repos: "The PEGR-Galaxy communication scripts are available at https://github.com/CEGRcode/pegr-galaxy_tools and the PEGR NGS pipeline scripts are available at https://github.com/CEGRcode/pegr-ngs_pipeline under the MIT license." There must be clear instructions for installing and enabling the advertised interconnectivity of PEGR and Galaxy. More reliance on Galaxy API could simplify a lot of the configuration currently needed within PEGR.

We agree with the reviewer that clear and detailed instructions for installing and configuring the PEGR-Galaxy communications tools is critical for user adoption. We have expanded the documentation available for tool installation: (https://github.com/CEGRcode/pegr-galaxy_tools)

We agree with the reviewer that increased integration with the Galaxy API would potentially decrease configuration requirements within PEGR. These upgrades are now also mentioned in our Discussion as a way forward for

multiple PEGR instances to communicate with a single Galaxy instance.

10. The PEGR-Galaxy communication scripts are available at https://github.com/CEGRcode/pegr-galaxy_tools

Listed in this manuscript (created 3 months ago):

```
$ git clone https://github.com/CEGRcode/pegr-galaxy_tools.git
Cloning into 'pegr-galaxy_tools'...
remote: Enumerating objects: 65, done.
remote: Counting objects: 100% (65/65), done.
remote: Compressing objects: 100% (24/24), done.
remote: Total 65 (delta 37), reused 65 (delta 37), pack-reused 0
Unpacking objects: 100% (65/65), 121.55 KiB | 313.00 KiB/s, done.
```

An unmentioned repository created 5 years ago, with additional Primary contributors not listed as authors nor listed under acknowledgements of current manuscript:

```
$ git clone https://github.com/seqcode/cegr-galaxy.git
Cloning into 'cegr-galaxy'...
remote: Enumerating objects: 1570, done.
remote: Counting objects: 100% (72/72), done.
remote: Compressing objects: 100% (22/22), done.
remote: Total 1570 (delta 49), reused 59 (delta 43), pack-reused 1498
Receiving objects: 100% (1570/1570), 1.65 MiB | 337.00 KiB/s, done.
Resolving deltas: 100% (928/928), done.
```

There are no differences between the tools in these separate repositories:

```
$ diff -r cegr-galaxy/tools/cegr_statistics/ pegr-galaxy_tools/tools/
```

It is not clear why a new repo was created 3 months ago with the tools manually copied over from a 5 year old repository without including attribution of the primary committer.

the PEGR NGS pipeline scripts are available at https://github.com/CEGRcode/pegr-ngs_pipeline

Listed in this manuscript (created 3 months ago):

```
$ git clone https://github.com/CEGRcode/pegr-ngs_pipeline.git
Cloning into 'pegr-ngs_pipeline'...
remote: Enumerating objects: 25, done.
remote: Counting objects: 100% (25/25), done.
remote: Compressing objects: 100% (24/24), done.
remote: Total 25 (delta 0), reused 25 (delta 0), pack-reused 0
Unpacking objects: 100% (25/25), 29.46 KiB | 1.40 MiB/s, done.
```

There is no difference between these "xEGR NGS pipeline scripts" (5 years ago, 3 months ago):

```
$ diff -r cegr-galaxy/scripts/ pegr-ngs_pipeline/scripts/
```

There are minimal differences between "cegr" and "pegr" sample configurations, such as changing naming and removing references to a home directory of user "gvk":

```
$ diff cegr-galaxy/config/cegr_config.ini.sample pegr-
ngs_pipeline/config/pegr_config.ini.sample
8,9c8,9
< ANALYSIS_PREP_LOG_FILE_DIR = /Users/gvk/work/git_workspace/cegr_galaxy/log
< API_KEY = 948f9b9e37c1ff5d39d6eala6cf13e46
---
> ANALYSIS_PREP_LOG_FILE_DIR = /pegr_galaxy/log
> API_KEY = XXXXXXXXXXXXXXXXXXXXXXXXXXXXXXXXXXXX
11,12c11,12
< BCL2FASTQ_BINARY = /Users/gvk/work/bcl2fastq_binary
< BCL2FASTQ_REPORT_DIR = /Users/gvk/work/bcl2fastq_reports
---
```

```

> BCL2FASTQ_BINARY = /path/to/bin/bcl2fastq_binary
> BCL2FASTQ_REPORT_DIR = /path/to/bin/bcl2fastq_reports
18c18
< # this enhancement was added manually to the ~/src/FastQValidator.cpp file:
---
> # this enhancement was added manually to the ~/src/FastQValidator.cpp file:
21c21
< FASTQ_VALIDATOR_BINARY = /Users/gvk/work/fastQValidator
---
> FASTQ_VALIDATOR_BINARY = /path/to/bin/fastQValidator
24c24
< GALAXY_HOME = /Users/gvk/work/git_workspace/galaxy
---
> GALAXY_HOME = /galaxy
26c26
< LIBRARY_PREP_DIR = /Users/gvk/work/git_workspace/cegr_galaxy/config/library_prep_dir
---
> LIBRARY_PREP_DIR = /pegr_galaxy/config/library_prep_dir
29c29
< PREP_VIRTUAL_ENV = /Users/gvk/work/git_workspace/cegr_galaxy/venv/bin/activate_this.py
---
> PREP_VIRTUAL_ENV = /pegr_galaxy/venv/bin/activate_this.py
31,32c31,32
< RAW_DATA_DIR = /Users/gvk/work/git_workspace/raw_data_dir
< RAW_DATA_LOGIN = nextseq@146.186.153.198
---
> RAW_DATA_DIR = /raw_data_dir
> RAW_DATA_LOGIN = nextseq@123.456.78.90
35c35
< REMOTE_RUN_INFO_FILE = /home/nextseq/NSQData_PughLab/cegr_run_info.txt
---
> REMOTE_RUN_INFO_FILE = /home/nextseq/NSQData_PughLab/pegr_run_info.txt
37,38c37,38
< REMOTE_WORKFLOW_CONFIG_DIR_NAME = cegr_config
< RUN_INFO_FILE = /Users/gvk/work/git_workspace/cegr_galaxy/config/cegr_run_info.txt
---
> REMOTE_WORKFLOW_CONFIG_DIR_NAME = pegr_config
> RUN_INFO_FILE = /pegr_galaxy/config/pegr_run_info.txt
40c40
< SAMPLE_SHEET = /Users/gvk/work/git_workspace/cegr_galaxy/config/cegr_sample_sheet.csv
---
> SAMPLE_SHEET = /pegr_galaxy/config/cegr_sample_sheet.csv
63,64c63,64
< SINGLE = single_001
< PAIRED = paired_001
---
> SINGLE = single_002
> PAIRED = paired_002
82,84d81
< NC003552 = /nfs/brubeck.bx.psu.edu/scratch5/galaxy-pugh/galaxy/tool-
data/NC003552/len/NC003552.len
< pa01 = /nfs/brubeck.bx.psu.edu/scratch5/galaxy-pugh/galaxy/tool-data/pa01/len/pa01.len
< pf25 = /nfs/brubeck.bx.psu.edu/scratch5/galaxy-pugh/galaxy/tool-data/pf25/len/pf25.len
86,88d82
< sp8 = /nfs/brubeck.bx.psu.edu/scratch5/galaxy-pugh/galaxy/tool-data/sp8/len/sp8.len
< # TODO: tair10.len is not available in this Galaxy instance
< # tair10 = /nfs/brubeck.bx.psu.edu/scratch5/galaxy-pugh/galaxy/tool-data/

```

It is not clear why a new repo was created 3 months ago with the scripts manually copied over from a 5 year old repository without including attribution of the primary committer.

We apologize for any confusion. The GitHub repo referenced by the reviewer was the original developmental repo, however over the course of the 6 years of the project, as a separate matter from the project, the NIH funded Principal Investigators for this project changed institutions years ago, which changed the distributions and control of the NIH funding. The PIs involved decided a

new repo would be the easiest mechanism for the move to the new institution. While the original attribution was maintained for this repo:

(https://github.com/CEGRcode/pegr-ngs_pipeline)

we neglected to include the attribution under the second repo generated:

(https://github.com/CEGRcode/pegr-galaxy_tools)

The lack of attribution during the repo change was an oversight and has now been rectified. We also note that the file path changes remarked upon by the reviewer were chosen for the purpose of emphasizing generalized file paths new users would need to customize for their own PEGR deployments. While we regret any confusion, the nature of this explanation goes far beyond what we feel should be reported in the manuscript.

(https://github.com/CEGRcode/pegr-galaxy_tools)

Acknowledgements

The authors thank Greg Von Kuster, Geoffrey Billy, Belinda Giardine, Abeer Almutairy, Hedgie Jo, and Pierce Chafflin for their user testing and feedback, code base tests, development code contributions and helpful discussions and feedback to this project. Computations for this research were performed on the Pennsylvania State University's Institute for Computational and Sciences' ROAR supercomputer.

11. It would be helpful if the authors could highlight the specific differences between "PEGR" and "CEGR" [my apologies if this is not the correct naming for this previously existing system], in particular the interconnectivity of the LIMS and analysis workflow systems (e.g. Galaxy).

We apologize for any confusion. However, as we noted above, this distinction is related to a change in institution that occurred over the life of the project. As noted in the GitHub, CEGR represents an organization 'Center for Eukaryotic Gene Regulation' and is composed of a number of PI's with like-minded research interests. The PEGR (Platform for EpiGenomic Research) software represents our combined efforts to address many of the outstanding challenges in rigorous and reproducible genomic research.

Reviewer #2: The paper introduces a web-based platform for sample tracking and project management of genomic experiments that integrates with bioinformatics analysis performed on Galaxy. The paper is well written and provides sound justification for the approach and good technical details, with several screenshots and diagrams that clarify how the system works.

1. An initial version of the platform was presented in 2020 on ACM Intl. Conference Proceeding Series, and although it is still probably early days, it would be nice if the authors could include some numbers on user uptake, or at least a description of its impact where it was deployed at their labs.

We agree with the reviewer that it would be of value to describe its current usage and impact on the research community. To date, PEGR has been adopted for beta-testing and production usage by the Pugh lab, Lai lab, Mahony lab, and Cornell Epigenomics Core. PEGR has supported the following publications directly and is cited as such within the acknowledgements. We have now modified the Introduction to better reflect the broad utility PEGR has provided to the labs that have adopted it.

To date, PEGR has been officially acknowledged in two papers generating thousands of genomic datasets and is actively utilized by multiple (Lai, Pugh, and Mahony) research labs and the Cornell Epigenomic Core Facility [31, 32].

In further support of open and accessible research, we have also registered a Research Resource Identifier (RRID) for PEGR in order to better track its impact in the future (RRID:SCR_021861).

When PEGR software is used as the platform providing the data, it can be cited using the RRID:SCR_021861.

2. I have tried without success to install PEGR following the instruction on GitHub (see suggestion below about providing a Docker image or demo server), so I'll have to base my comments on the description given in the paper.

We greatly appreciate the reviewer's suggestion to provide a Docker image to remove the need for a user to deploy and operation a production version of PEGR. A Docker instance of the latest version of PEGR is now available through the Docker Hub:

<https://hub.docker.com/repository/docker/dshao/pegr>

3. Integrating platforms like LIMS and workflow management systems is a complex undertaking (especially for an open source project), where there is a risk of creating a platform that is too complex or not good enough (in terms of features or usability) for the single tasks. In this respect, I support the authors' approach of not reinventing the wheel for the analysis subsystem, although the way in which Galaxy tools are integrated seems half-baked (see detailed comment below).

We agree with the Reviewer that PEGR-Galaxy communication tools are not fully optimized, and this was an issue also raised by Reviewer 1. However, in our current design, by requiring Galaxy to POST to PEGR, this dramatically reduces the overhead inherent in PEGR continually polling Galaxy for job status. This would also to some extent reproduce Galaxy workflow functionality within PEGR, which as the Reviewer notes would be an unnecessary reinvention of the wheel. In support of greater generalization and also noted

before, we now provide a generalized tool which communicates from nearly any given tool in Galaxy to PEGR, while still sending critical required information about the status of the Galaxy tool and analyzed dataset: (https://github.com/CEGRcode/pegr-qalaxy_tools/commit/06f836cf2fb46e452119ad8f6c45650cb6a2e370) **We have also expanded the Discussion to note that future development will focus on better optimization and generalization of PEGR-Galaxy tool communication.**

We also believe that given the prominence of many internationally funded Galaxy instances (e.g., <https://usegalaxy.org/>, <https://usegalaxy.eu/>), a key future upgrade will be to enable multiple PEGR instances to communicate with multiple Galaxy instances in a full many-to-many relationship. This will enable researchers to directly benefit from well-funded bioinformatic rigor and reproducibility initiatives by reducing the overhead required for smaller groups to run their own Galaxy instances.

In conclusion, the paper and the software address the important issue of extending reproducibility from the wet lab up to the final report of a bioinformatic analysis, and I hope the authors succeed in popularising PEGR and continue its development.

Comments:

- Background, 1st paragraph: references 4-11 seem very general and not particularly relevant to the reproducibility issues described.

We appreciate the reviewer's comments and have updated the references to include additional publications directly related to issues in experimental reproducibility including: Devailly et al. discussing lack of reproducibility in ENCODE ChIP-seq data (PMID: 26619763), Yardimci et al. discussing challenges in Hi-C reproducibility (PMID: 30890172), ENCODE attempts to address reproducible data (PMID: 22955991), and other relevant citations.

- Page 5, 1st paragraph: "To prevent disorganization ..., ItemType's are organized into an ItemTypeCategory" - maybe replace "organized" with "grouped". Also, is this a proper hierarchical structure (with ItemCategory's grouped in a higher ItemCatoregory)?

Fixed. ItemTypeCategory's are not currently designed to be hierarchical, however we can add that functionality if users find a greater need for better hierarchical organization.

- Page 5: Does PEGR comw with some predefined ItemType's? Would a AppStore-like website be helpful for sharing definitions among PEGR admins?

PEGR does come with a variety of pre-defined ItemType's that are designed to support ChIP-seq and RNA-seq genomic experiments. We like the idea of a AppStore-style interface and would be interested in supporting that as an extended feature system in the future. In the short-term we have provided a mechanism by which a user can upload a CSV-file containing a large number of ItemType's. This provides a mechanism for a user to initialize a PEGR instance customized for their particular use without requiring them to manually input each item in through the webform.

- Figure 3 caption: "(E) 'ProtocolGroup' is accessible through the Admin console" - Why can ProtocolGroup's can be created only by admins? This seems something that can become a bottleneck.

This design consideration was strongly suggested by Dr. Frank Pugh who provided

critical UX insight based on his extensive biochemistry training experience (>30 years). A common issue in laboratories that generate novel genomic assays is the production of a wide-range of variant assays that never see the light of day. While these approaches should be thoroughly documented, they do not represent a best-practice protocol for the general lab. This was the rationale for why a PEGR user is able to initialize and execute any novel protocol and experiment but is unable to formalize it into a ProtocolGroup visualized by all lab members. A ProtocolGroup represents a thoroughly vetted workflow that has passed some form of review to be adopted as a general laboratory protocol. We have now expanded the Protocol section to better explain this.

ProtocolGroup's are initialized through the Admin console. This design consideration requires a ProtocolGroup to be thoroughly vetted by a PEGR administrator (i.e., Principal investigator, lab manager) before it can be accessed and used by the entire group. While users are still able to construct and initialize any individual Protocol they desire, this produces an intentional pause-point in developing novel assays which requires users (i.e., graduate students) to reflect on their experimental design and discuss with a relevant senior scientist.

- Page 9, 3rd paragraph: "A traced sample can be added to an experiment using either the web-interface or the QR barcode system" - Can samples be added in advance (before creating a new Experiment? There is a "Samples" button in the top menu.

This point was also raised by Reviewer 1 and has been addressed above. In short, we believe that a clear delineation in when a Sample comes into existence occurs when the experiment is performed. This functionality reflects real life wet-bench best practices and we believe is relevant to the PEGR user experience.

A typical lab process is to generate common laboratory reagent stocks (e.g., wash buffers) that are used multiple times across many different downstream experiments. However, more complicated experimental setups like ChIP-seq, involve a 'traced' sample which moves through multiple sequential experiments and combines with different reagents as it transitions through product states (e.g., sonicated chromatin converts to DNA library). A 'traced' sample typically begins as a 'BioSample' in PEGR. The 'BioSample' is assigned a unique 'Sample' ID within the PEGR database the moment it is added to an Experiment. This provides a clear delineation in the creation of new Samples in PEGR and helps to prevent users from initializing any number of theoretical Samples that are unlinked to any Experiment. This functionality mirrors the best practices of a standard laboratory notebook. As lab notebooks are not designed to record proposed experiments, but only the record of a performed Experiment, this logic is consistent with standard biochemical wet-bench practices. Importantly in the case of traced samples, PEGR can display all the states that a sample has transitioned through allowing for full experimental history tracking. A traced sample can be added to an experiment using either the web-interface or the QR barcode system (Figure 4E). Importantly, PEGR allows multiple samples to be attached to a single protocol. This enables the operator to process multiple samples in a batch while only needing to enter the related information once (e.g., when performing ChIP-seq on 8 samples in parallel).

- Page 24, Availability of data and materials: I'd suggest to also make PEGR available as a Docker image and/or a demo server. I've tried to run it following the instructions at <https://github.com/seqcode/pegr> but ended with a long page full of Java exceptions.

We agree wholeheartedly and note this issue was raised by Reviewer 1 as well. As mentioned above, we now provide a Docker instance of PEGR for deployment without requiring a user to install the full suite of software dependencies.

(<https://hub.docker.com/repository/docker/dshao/pegr>)

- Page 20: "When an analysis step finishes, its output data will be posted to PEGR immediately." - It's not clear how this happens, was Galaxy modified to initiate a HTTP POST request towards the PEGR RESTful API after a tool is run, or is it the tool's responsibility? By looking at https://github.com/CEGRcode/pegr-galaxy_tools it seems it is

the latter, in fact each Galaxy tool published there sends its output to PEGR autonomously. If that is the case, this is a serious limitation to the usefulness of PEGR, because users wouldn't be able to directly make use of the thousands of Galaxy tools available on the Galaxy ToolShed.

This issue was raised with Reviewer 1 as well. We agree that relying on individual autonomous tools dramatically reduces the plug-and-play nature of Galaxy-PEGR communications. However, we note that the majority of tools required to communicate the progress of a workflow to PEGR can likely function as a simple Boolean call of pass/fail. This would require only a far smaller pool of tools to require custom autonomous POST requests to PEGR with results tailored to the specialized analysis. We have now made a generalized PEGR-Galaxy communication tool available on the Git repo:

(https://github.com/CEGRcode/pegr-galaxy_tools/commit/06f836cf2fb46e452119ad8f6c45650cb6a2e370)

Minor comments:

- Background, 1st paragraph: "What is needed is systematic metadata capture..." - This sentence could be made a bit more absolute, something like "One way to tackle these issues is to apply systematic metadata capture..."

Edited as follows:

One method to address these issues is to apply systematic metadata capture and management software that is tailored to (epi)genomic data collection.

- Background, 3rd paragraph: <https://doi.org/10.1093/bioinformatics/btt115> is an (abandoned) attempt to extend Galaxy with LIMS functionalities that may be worth to mention.

Edited as follows:

To our knowledge, there are no free open-source platforms in active development that manage entire experimental pipelines, from wet-bench experiments to bioinformatic analyses [23].

- Background, 4th paragraph: "Galaxy.org" -> "Galaxy"

Fixed

- Page 7, last paragraph: "organize the variety of protocols that often come sequentially in a pipeline": do protocols have to come in a sequence? I think it should be possible to execute some steps in parallel, so maybe "organize the variety of protocols that often compose an experiment"

Edited as suggested:

Similar to how ItemTypeCategory is used to organize the wide variety of ItemTypes in the 'Inventory', Protocol Groups are used to consolidate and organize the variety of protocols that often compose an experiment (Figure 3E).

- Page 9, 1st paragraph: "The PEGR 'Experiment' interface is designed to track and maintain the relational links between reagents, protocols, and the resulting end products" - I think that "samples" should be listed here even if they are formally introduced some lines below.

Edited as suggested:

The PEGR 'Experiment' interface is designed to track and maintain the relational links between reagents (i.e., 'Inventory'), protocols (i.e., 'Protocol'), and the resulting end products (i.e., 'Samples').

- Figure 4 caption: "...the effect of electing a Protocol Group" - "electing" -> "selecting"

Fixed

- Page 11, 2nd paragraph: "...XML-wrapper python scripts which send a JSON file to PEGR in a standard POST request." -> "...XML-wrappers for Python scripts which send a JSON file to PEGR RESTful API in a standard HTTP POST request."

Edited as suggested:

Galaxy workflows designed to communicate with PEGR contain simple XML-wrappers for Python scripts which send a JSON file to PEGR RESTful API in a standard HTTP POST request.

- Figure 6 caption: "...the users affiliated projects..." -> "...the user's affiliated projects..."

Fixed

- Page 16, 1st paragraph: "It supports reproducibility... management [22, 26]" - A bit unreadable, I would rewrite as "It supports scientific data management by tracking samples from the very first step of sample preparation to the end of bioinformatics analysis and data reporting thus supporting the FAIR principles [22] and the reproducibility goals of the Galaxy platform [26]."

Edited as suggested:

It supports scientific data management by tracking samples from the very first step of sample preparation to the end of bioinformatics analysis and data reporting, thus supporting the FAIR principles and the reproducibility goals of the Galaxy platform [29, 33].

- Supplemental Figure 1: It would be useful to include or link the source file used to generate this figure, the diagrams are too small to be useful/readable.

We apologize to the Reviewer for the illegibility of Supplemental Figure 1. We have also added the higher resolution schema for the pegrDB directly into the README on the Git repo now:

(<https://github.com/seqcode/pegr/commit/dfd35f34ab6ee8a1e0f4d22b568fd6885dc1d2ba>)

Second round of review

Reviewer 1

The authors have largely addressed my concerns from the original submission with updates to both the manuscript and the software.

Starting with a fresh Ubuntu 20.04.3 virtual machine running under VirtualBox that was updated and had required packages installed. I was able to install and run v0.3.0 of PEGR (<https://github.com/seqcode/pegr/releases/tag/v0.3.0>).

Installation and setup was pretty straight forward. Much of the difficulty in using the software is due to the lack of an initial walkthrough in usage from both an admin and user perspective. This could be done using screencast videos or detailed text with pictures. There is a “guide” available within the PEGR interface, but it is very minimal. There is a wiki available within github, but this is minimal as well. There is some overlap between them, but there is no extensive user/admin manual for actually using the software.

I would recommend that the authors find some volunteers that have never used PEGR previously, give them a freshly installed and configured server, the admin username and password, tell them that there is a “guide” option in the top of the PEGR software and a link to the wiki on github. Will a user be able to get started? Will they be able to add (bio)samples (and other inventory) and be able to start/progress an experiment? Even with the pre-defined protocols already loaded this seems to be a very difficult task with the current documentation.

README.md should be updated to match the latest released version (it still points to v0.1.0).

The addition of a generalized Galaxy tool to report back to a PEGR instance is good. It currently requires the configuration of a single static ini file inside of the tool directory; it might be helpful to additionally allow the configuration of the PEGR instance to be completed within the tool interface itself – then a workflow can be dynamically configured (either ‘hardcoded’ within the specific workflow, or at runtime of the workflow) by the Galaxy user to point to the PEGR server of interest along with a PEGR api key. This would possibly make good progress on enabling a many-to-many PEGR-Galaxy relationship. One could also add a repeat block that contains 2 text inputs to the Galaxy tool, representing key, value pairs to enable additional information to be sent back from Galaxy to PEGR as desired (again either defined in the workflow, or specified at workflow runtime).

We kindly thank the reviewer for their critical assessments. Our responses to the issues raised are **written in bold**.

Reviewer #1: The authors have largely addressed my concerns from the original submission with updates to both the manuscript and the software.

Starting with a fresh Ubuntu 20.04.3 virtual machine running under VirtualBox that was updated and had required packages installed. I was able to install and run v0.3.0 of PEGR (<https://github.com/seqcode/pegr/releases/tag/v0.3.0>).

Installation and setup was pretty straight forward. Much of the difficulty in using the software is due to the lack of an initial walkthrough in usage from both an admin and user perspective. This could be done using screencast videos or detailed text with pictures. There is a "guide" available within the PEGR interface, but it is very minimal. There is a wiki available within github, but this is minimal as well. There is some overlap between them, but there is no extensive user/admin manual for actually using the software.

I would recommend that the authors find some volunteers that have never used PEGR previously, give them a freshly installed and configured server, the admin username and password, tell them that there is a "guide" option in the top of the PEGR software and a link to the wiki on github. Will a user be able to get started? Will they be able to add (bio)samples (and other inventory) and be able to start/progress an experiment? Even with the pre-defined protocols already loaded this seems to be a very difficult task with the current documentation.

We have expanded the PEGR guide and the corresponding wiki with increased documentation for getting started with PEGR.

(<https://github.com/seqcode/pegr/commit/317c0dfe8ce392b4dd0d0015f6e85ae40b20da2c>)

(<https://github.com/seqcode/pegr/wiki/What-is-PEGR%3F>)

README.md should be updated to match the latest released version (it still points to v0.1.0).

Fixed

(<https://github.com/seqcode/pegr/commit/317c0dfe8ce392b4dd0d0015f6e85ae40b20da2c>)

The addition of a generalized Galaxy tool to report back to a PEGR instance is good. It currently requires the configuration of a single static ini file inside of the tool directory; it might be helpful to additionally allow the configuration of the PEGR instance to be completed within the tool interface itself – then a workflow can be dynamically configured (either 'hardcoded' within the specific workflow, or at runtime of the workflow) by the Galaxy user to point to the PEGR server of interest along with a PEGR api key. This would possibly make good progress on enabling a many-to-many PEGR-Galaxy relationship. One could also add a repeat block that contains 2 text inputs to the Galaxy tool, representing key, value pairs to enable additional information to be sent back from Galaxy to PEGR as desired (again either defined in the workflow, or specified at workflow runtime).

We agree with the reviewer that updating the communication tool would allow for substantially increased flexibility including a many-to-many relationship between PEGR and Galaxy. We have now updated the tool to accept any arbitrary API key and URL from within the Galaxy interface which will allow diverse locations for the JSON POST request.

(https://github.com/CEGRcode/pegr-galaxy_tools/commit/27efeebbbdac7d22fbd3f8b72da69cdd36e6277c)